# THE ACT OF REMEMBERING: A STUDY IN PARTIALLY OBSERVABLE REINFORCEMENT LEARNING

## ABSTRACT

Partial observability remains a major challenge for reinforcement learning (RL). In fully observable environments it is sufficient for RL agents to learn memoryless policies. However, some form of memory is necessary when RL agents are faced with partial observability. In this paper we study a lightweight approach: we augment the environment with an external memory and additional actions to control what, if anything, is written to the memory. At every step, the current memory state is part of the agent's observation, and the agent selects a tuple of actions: one action that modifies the environment and another that modifies the memory. When the external memory is sufficiently expressive, optimal memoryless policies yield globally optimal solutions. We develop the theory for memory-augmented environments and formalize the RL problem. Previous attempts to use external memory in the form of binary memory have produced poor results in practice. We propose and experimentally evaluate alternative forms of k-size buffer memory where the agent can decide to remember observations by pushing (or not) them into the buffer. Our memories are simple to implement and outperform binary and LSTM-based memories in well-established partially observable domains.

## 1 INTRODUCTION

Reinforcement Learning (RL) agents learn policies (i.e., mappings from observations to actions) by interacting with an environment. RL agents usually learn *memoryless* policies, which are policies that only consider the last observation when selecting the next action. In fully observable environments, learning memoryless policies is an effective strategy. However, RL methods often struggle when the environment is partially observable. Indeed, partial observability is one of the main challenges to applying RL in real-world settings (Dulac-Arnold et al., 2019).

When faced with partially observable environments, RL agents require some form of memory to learn optimal behaviours. This is usually accomplished using k-order memories (Mnih et al., 2015), recurrent networks (Hausknecht & Stone, 2015), or memory-augmented networks (Oh et al., 2016).

In this paper, we study a lightweight alternative approach to tackle partially observability in RL. The approach consists of providing the agent with an external memory and extra actions to control it (as shown in Figure 1). The resulting RL problem is still partially observable, but if the external memory is sufficiently expressive, then optimal memoryless policies will also yield globally optimal solutions. Previous works that explored this idea using external *binary* or *continuous* memories produced poor results with standard RL methods (Peshkin et al., 1999; Zhang et al., 2016). Our work shows that the main issue is with the type of memory they were using, and that RL agents are capable of learning effective strategies to utilize external memories when structured appropriately.

In what follows, we

- formalize the RL problem in the context of memory-augmented environments and study the theory behind memoryless policies that jointly decide what to do and what to remember;
- propose two novel forms of external memory called O$k$ and OA$k$. These k-size buffer memories generalize k-order memories by letting the agent (learn to) *decide* whether to push the current observation into the memory buffer or not;
- empirically evaluate O$k$ and OA$k$ relative to previously proposed binary (B$k$), k-order (K$k$), and LSTM memories (the most widely used approach for partially observable RL).

Figure 1: A diagram of a Memory-Augmented Environment.

Results show that $O_k$ and $OA_k$ memories are usually more sample efficient than LSTM memories, while being faster to train and trivial to integrate with off-the-shelf RL methods. We therefore advocate for the adoption of $O_k$ and $OA_k$ memories for partially observable RL problems. We end with a discussion of limitations of $O_k$ and $OA_k$ and interesting avenues for future work.

## 2 PRELIMINARIES

RL agents learn how to act by interacting with an environment. Often these environments are modelled as a *Markov Decision Process (MDP)*. An MDP is a tuple $\mathcal{M} = \langle S, A, R, p, \gamma, \mu \rangle$, where $S$ is a finite set of *states*, $A$ is a finite set of *actions*, $R$ is the finite set of possible rewards, $p(s', r|s, a)$ defines the *dynamics* of the MDP, $\gamma$ is the *discount factor*, and $\mu$ is the *initial state distribution*. The interaction is usually divided into *episodes*. At the beginning of an episode, the environment is set to an initial state $s_0$, sampled from $\mu$. Then, at time step $t$, the agent observes the current state $s_t \in S$ and executes an action $a_t \in A$. In response, the environment returns the next state $s_{t+1}$ and immediate reward $r_t$ sampled from $p(s_{t+1}, r_t|s_t, a_t)$. The process then repeats until the end of the episode (when a new episode will begin) or potentially keep going for ever in *non-episodic* MDPs.

Agents select actions according to a *policy* $\pi(a|s)$—which is a probability distribution from states to actions. The *prediction* task is to estimate how "good" a policy is, where the policy is evaluated according to the *expected discounted return* in any state. This can be done by estimating the *action-value function* $q_\pi$ of policy $\pi$, where $q_\pi(s, a)$ represents the expected discounted return when executing action $a$ in state $s$ and following $\pi$ thereafter. Formally,

$$q_\pi(s, a) = \mathbb{E}_\pi \left[ \sum_{k=0}^{\infty} \gamma^k r_{t+k} \middle| S_t = s, A_t = a \right],$$

where $\mathbb{E}_\pi[\cdot]$ denotes the expected value of a random variable given that the agent follows policy $\pi$, and $t$ is any time step. $q_\pi$ is usually estimated using *Monte Carlo* samples (Barto & Duff, 1994) or *TD* methods (Sutton, 1988). The *control* task involves finding the *optimal* policy $\pi^*$. This is the policy that maximizes the expected discounted return in every state. To do so, most RL methods rely on the *policy improvement* theorem, which we discuss in Section 5.

We use a *Partially Observable Markov Decision Process (POMDP)* formulation to model partial observability. A POMDP is a tuple $\mathcal{P} = \langle S, O, A, R, p, \omega, \gamma, \mu \rangle$, where $S$, $A$, $R$, $p$, $\gamma$, and $\mu$ are as in the MDP above, $O$ is a finite set of *observations*, and $\omega(o|s)$ is the *observation probability distribution*. Interacting with a POMDP is similar to an MDP. The environment starts from a sampled initial state $s_0 \sim \mu$. At time step $t$, the agent is in state $s_t \in S$, executes an action $a_t \in A$, receives an immediate reward $r_t$, and moves to $s_{t+1}$ according to $p(s_{t+1}, r_t|s_t, a_t)$. However, the agent does not observe $s_t$ directly. Instead, the agent observes $o_t \in O$, which is linked to $s_t$ via $\omega(o_t|s_t)$.

## 3 RELATED WORK

Early attempts to perform RL in partially observable domains focused on learning memoryless policies. Jaakkola et al. (1995) identified an RL algorithm that was guaranteed to converge to locally optimal memoryless policies, and similar guarantees have been given in the POMDP literature (Li et al., 2011). Unfortunately, Singh et al. (1994) showed that an optimal memoryless policy $\pi^*(a_t|o_t)$ can be arbitrarily worse than the optimal history-based policy $\pi^*(a_t|o_0, a_0, \ldots, o_t)$ for POMDPs.

Different approaches have been proposed to learn history-based policies using some form of state-approximation technique. For example, model-based RL methods learn a state representation of

histories that enables Markovian prediction of future observations, rewards, or expected returns, and then learns policies over that representation (McCallum, 1996; Littman et al., 2002; Poupart & Vlassis, 2008; Doshi-Velez et al., 2013; Ghavamzadeh et al., 2015; Zhang et al., 2019; Toro Icarte et al., 2019). The focus of our work is on model-free methods, which are the state of the art for solving partially observable problems from low-level inputs (such as images). In model-free RL, history-based policies are approximated using recurrent neural networks (Hausknecht & Stone, 2015; Mnih et al., 2016; Wang et al., 2016; Jaderberg et al., 2016), or some form of memory-augmented neural network (Oh et al., 2016; Parisotto & Salakhutdinov, 2017; Khan et al., 2017; Hung et al., 2018). They are usually trained using policy gradient methods. These approaches are computationally expensive because they require the backpropagation of gradients through the history of observations and actions for learning history-based policies. In comparison, our approach is much more lightweight – being faster to train than LSTMs and generally having better sample complexity.

We note that it is possible to learn memoryless policies that optimally solve POMDPs. The trick is to give the agent a (large enough) memory and extra actions to write to it. From the agent's perspective, it learns a standard memoryless policy from observations to actions, but the observations now include the state of the memory, and the actions include options for how to alter the memory. The main purpose of our work is to resurrect this simple idea by understanding why previous work were unable to make it work. We also proposed a unified framework to study agents with external memories and two novel memories that outperform existing forms of external memories.

Concretely, the idea of providing some form of external memory to an agent and actions to modify it goes back to Littman (1993), who discussed a hypothetical agent that could learn to control an external binary memory in support of solving partially observable tasks. Peshkin et al. (1999) reported the first empirical results using tabular RL to learn memoryless policies over such binary memories. While the results were promising in some small environments that required only one bit of external memory to be solved, they did not scale to more complex domains. After Peshkin et al. (1999), there was not much work trying to push this line of research forward. We believe that the reason is that RL agents cannot reliably learn to control binary memories (as shown in our results). That said, there is one recent work that has further explored the idea of modifying external memories using actions. Zhang et al. (2016) proposed to use continuous memories, where each element in the array was a floating point number, instead of binary memories. However, they learned the memoryless policies using imitation learning and pointed out that standard RL methods did not work because the reward signal was insufficient supervision for the agent to understand how to appropriately control the memory. One contribution of our work is to advance our understanding of methods that provide external memory to standard RL agents, and to show that they can work well in practice.

Our work is also related to neural Turing machines (NTM). The idea behind NTM is to provide an external memory to neural networks which they can write to and read from (Graves et al., 2014). All their components are differentiable and, hence, they can be trained end-to-end using gradient descent and a training set of input and output examples (i.e., they solve a supervised learning problem). Zaremba & Sutskever (2015) proposed a variation of NTMs where they used the Reinforce algorithm to control how to move the head that reads and write over a memory tape – which can be seen as a case where an RL agent learns to (partially) control an external memory. That said, their overall system still solves a supervised learning problem as it requires the supervision coming from input and output examples to train the rest of the components in the NTM.

## 4 AGENTS WITH EXTERNAL MEMORY

In this section, we formally define what it means to provide external memory to an agent, and describe several forms of external memory. We will use the following problem to aid explanation:

**Example 4.1** (the gravity domain (Toro Icarte et al., 2019)). *The* gravity domain*, shown in Figure 2, consists of an agent (purple triangle), a cookie, and a button. The agent can move in the four cardinal directions and receives a reward of 1 when it eats the cookie. Doing so ends the episode. There is an external force pulling the agent down—i.e., the outcome of the "*move-up*" action is a downward movement with probability 0.9—which can be turned off (or back on) by pressing the button. Every episode begins with the agent in the bottom left corner and the external force on.*

The optimal policy for this problem is to first press the button and then to go to the cookie. Since the agent cannot observe the force, optimal behaviour requires memory of the state of the button,

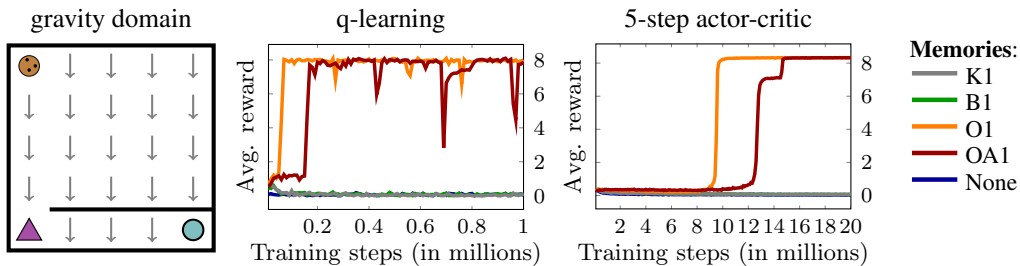

Figure 2: Experiments in the gravity domain. We reported the avg. reward per 100 steps.

meaning that no memoryless policies can solve this problem optimally. However, suppose that the agent was given a single bit that they could write to on every step using the special actions `write-1` and `write-0`. This memory can then be used to record the state of the button, and so an optimal memoryless policy for this augmented problem will optimally solve the gravity domain.

Figure 1 shows a generalization of this idea. From the agent's perspective, they are, as usual, performing actions in an environment and receiving observations and rewards in return. However, they are now interacting with a *memory-augmented environment*—which consists of a *sub-environment* (i.e., the original POMDP environment) and a *memory*. The memory receives an action $w$ (selected by the agent) and local information coming from the sub-environment $(o, a, r, o')$ to update its internal state to $m'$. We formalize these external memory modules as follows:

**Definition 4.1** (external memories). Let $\mathcal{P} = \langle S, O, A, R, p, \omega, \gamma, \mu \rangle$ be a POMDP. An external memory for $\mathcal{P}$ is a tuple $\mathcal{M}_{\mathcal{P}} = \langle M, W, \Gamma, \eta \rangle$, where $M$ is a finite set of *memory-states*, $W$ is a finite set of *memory-writing actions*, $\Gamma(m'|m, w, o, a, r, o')$ is the *memory-writing distribution*, and $\eta$ is the *initial memory-state distribution*.

An external memory module defines the set of possible memory configurations ($M$) and how the agent can manipulate that memory ($W$ and $\Gamma$). In the one-bit example for the gravity domain, $M$ consists of the two possible values of the bit (0 or 1), $W$ consists of the two possible write options of the bit (0 or 1), and the memory-writing distribution $\Gamma$ updates the bit of memory to 0 or 1 depending on which action was selected. We now define a memory-augmented environment as follows:

**Definition 4.2** (memory-augmented environments). A memory-augmented environment is a tuple $\mathcal{E} = \langle \mathcal{P}, \mathcal{M}_{\mathcal{P}} \rangle$ where $\mathcal{P}$ is a POMDP and $\mathcal{M}_{\mathcal{P}}$ is an external memory for $\mathcal{P}$.

The interaction between an agent and a memory-augmented environment $\mathcal{E} = \langle \mathcal{P}, \mathcal{M}_{\mathcal{P}} \rangle$ is the same as with the original environment, just with an augmented observation and action space. At the beginning of each episode, an initial state $s_0$, observation $o_0$, and memory state $m_0$, are sampled according to $s_0 \sim \mu$, $o_0 \sim \omega(o_0|s_0)$, and $m_0 \sim \eta$, respectively. At time step $t$, the agent observes $\bar{o}_t = \langle o_t, m_t \rangle$ and executes an action $\bar{a}_t = \langle a_t, w_t \rangle \in A \times W$ in $\mathcal{E}$. Consequently, the sub-environment samples an immediate reward $r_t$ and the next state $s_{t+1}$ according to $p(s_{t+1}, r_t|s_t, a_t)$. The sub-environment also samples the next observation $o_{t+1} \sim \omega(o_{t+1}|s_{t+1})$. The memory state is then updated to $m_{t+1}$ according to $\Gamma(m_{t+1}|m_t, w_t, o_t, a_t, r_t, o_{t+1})$. Finally, the agent receives the immediate reward $r_t$ and the next observation $\bar{o}_{t+1} = \langle o_{t+1}, m_{t+1} \rangle$, and the process repeats.

Any standard RL algorithm can be used to find a memoryless policy for a given memory-augmented environment $\mathcal{E} = \langle \mathcal{P}, \mathcal{M}_{\mathcal{P}} \rangle$. We note that the optimal memoryless policy for $\mathcal{E}$ must be at least as good as the optimal memoryless policy for the original POMDP $\mathcal{P}$. This is because $\mathcal{E}$ and $\mathcal{P}$ share a reward function, and the agent can always choose to ignore the memory. That said, if the external memory module $\mathcal{M}_{\mathcal{P}}$ is "expressive enough," then optimal memoryless policies for $\langle \mathcal{P}, \mathcal{M}_{\mathcal{P}} \rangle$ will be just as good as the optimal policy for $\mathcal{P}$. This is shown formally in Appendix A.2.

### 4.1 EXTERNAL MEMORY MODULES

Let us now consider several examples of external memory modules. We begin by showing how binary memories (Littman, 1993; 1994; Peshkin et al., 1999) can be expressed using this formalism. We use the notation $Bk$ to refer to a binary memory of $k$ bits:

**Definition 4.3** (B$k$ memories). Let $\mathcal{P} = \langle S, O, A, R, p, \omega, \gamma, \mu \rangle$ be a POMDP. A *Bk memory* for $\mathcal{P}$ is a $k$-bit external memory $\mathcal{M}_{\mathcal{P}} = \langle M, W, \Gamma, \eta \rangle$, where $M = \{0,1\}^k$, $W = \{0,1\}^k$, $\eta(0^k) = 1$ (zero otherwise), and $\Gamma(m'|m, w, o, a, r, o') = 1$ if and only if $m' = w$ (zero otherwise).

B$k$ memories are especially attractive given how flexible and expressive they are. Unfortunately, learning to control B$k$ memories is difficult. Figure 2 shows the performance of tabular q-learning (Watkins & Dayan, 1992) and 5-step actor-critic (Grondman et al., 2012) in the gravity domain using different types of external memories. In the figure, *None* represents not using any external memory, and K1, O1, and OA1 are explained below. Notice that the agent using q-learning or 5-step actor-critic was unable to learn how to use the B1 memory to consistently solve the gravity domain.

There are two main problems with B$k$ memories. First, the action space grows exponentially with $k$. Second, B$k$ memories can be *too* flexible in that the agent can modify the memory arbitrarily and irrespective of what has actually happened, and thereby completely alter what the agent believes about its current situation. For example, recall that in the gravity domain, the agent should use the memory to record whether gravity is on (0) or off (1). However, if the agent incorrectly decides to record that the gravity is off prematurely (i.e., before touching the button), it will believe it has transitioned from a state with low expected reward (where it first has to go to the button) to a state with high expected reward (where the agent wrongly believes that it can go directly to the cookie without any opposition from gravity). This can lead to an unstable learning process, as shown below.

The main motivation behind our proposed O$k$ memories is to alleviate these issues. O$k$ memories are a generalization of *k-order* memories, which are buffers of a fixed size that contain the last k observations. We refer to k-order memories as *Kk* memories, where the second '$k$' indicates the size of the buffer. We formally describe them as external memories in Appendix A.3. Note that K1 represents a 2-order memory since actions are taken over $\langle o, m \rangle$. The main disadvantage of k-order memories is that they do not allow the agent to remember events that occurred more than k steps in the past. O$k$ memories solve this issue by letting the agent decide whether to push the current observation into the k-order buffer or not. Since the agent can only push into the buffer observations that did occur, O$k$ memories are unable to imagine events that have not yet happened.

**Definition 4.4** (O$k$ memories). Let $\mathcal{P} = \langle S, O, A, R, p, \omega, \gamma, \mu \rangle$ be a POMDP. An *Ok memory* for $\mathcal{P}$ is a memory buffer (of size $k$) $\mathcal{M}_{\mathcal{P}} = \langle M, W, \Gamma, \eta \rangle$, where $M = (O \cup \{\emptyset\})^k$, $W = \{\top, \bot\}$, $\eta(\emptyset^k) = 1$ (zero otherwise), and $\Gamma(m'|m, w, o, a, r, o') = 1$ if $w = \bot$ and $m' = m$, or $w = \top$, $m = \langle o^1, o^2, \cdots, o^k \rangle$, and $m' = \langle o^2, \cdots, o^k, o \rangle$ (zero otherwise).

O$k$ memories have strong empirical performance in the gravity domain (see Figure 2), outperforming B1 and K1. That said, O$k$ memories are insufficient in domains where the history of actions matters. For such domains, we propose *OAk* memories. An OA$k$ memory is similar to an O$k$ memory but when the agent chooses to push to its buffer, the information includes the current observation and the action that is executed in the sub-environment. OA$k$ memories are defined in Appendix A.3.

We note that optimal memoryless policies over B$k$, OA$k$, O$k$, and K$k$ will optimally solve the original POMDP for some value of $k$, under some assumptions. This is shown in Appendix A.4.

## 5 LEARNING POLICIES IN MEMORY-AUGMENTED ENVIRONMENTS

The objective of this section is to understand the theory behind learning memoryless policies over memory-augmented environments and to provide insights into why O$k$ and OA$k$ memories tend to perform better than B$k$ memories. We begin with the following example.

**Example 5.1** (a recall task). *The recall task is a partially observable environment with only one possible observation, o (i.e., all states appear the same), and three actions, $a_1$, $a_2$, and $a_3$. The episode ends after performing three actions. If the agent executes actions $a_1$, $a_2$, and $a_3$ (in that order), it gets a reward of 1; otherwise it gets a reward of 0.*

The purpose of the recall task is to show that even if a memoryless policy for a memory-augmented environment is globally optimal, the memory-augmented environment itself might not be an MDP. Figure 3 shows a transition diagram for the recall task using an OA1 memory. Since the observation is always the same, the different states that the agent encounters only differ by the state in the memory. In the diagram, nodes represent the memory states and the transitions show how the memory is updated by the agent's actions. Note that node $i$ represents that the memory buffer contains $\langle o, a_i \rangle$ and that the buffer starts empty ($\emptyset$). For the action labels, the first number indicates the action

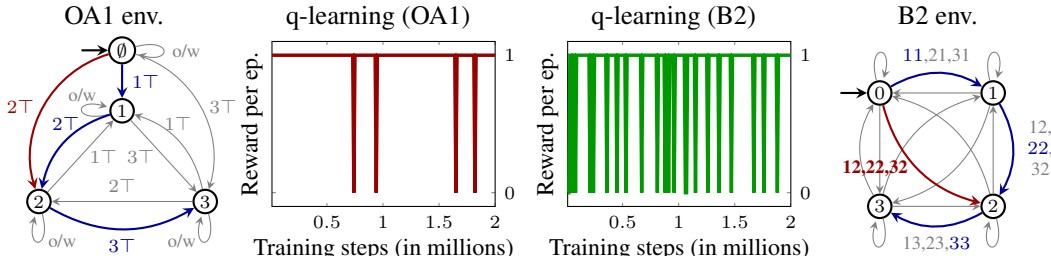

Figure 3: Performance of a greedy policy every $10,000$ training steps in the recall task.

number in the sub-environment (1, 2, or 3). The second character represents the memory action. For instance, $2\top$ represents that the agent executed $a_2$ in the sub-environment and *saved* that action into the memory buffer. The label o/w stands for otherwise. The blue arrows show a deterministic memoryless policy that optimally solves this problem. That is, execute $1\top$, then $2\top$, and finally $3\top$.

Notice that this memory-augmented environment has a memoryless policy that is optimal for the original POMDP, but it is not an MDP. The reason is that the reward given by the bottom transition $3\top$ will be 0 or 1 depending on the history. If the agent follows the blue path, it gets a reward of one. If the agent follows the red arrow, it gets a reward of zero. Something similar occurs when using B2 memory, which is the smallest B$k$ memory that can encode an optimal policy for this task. This "non-Markovianess" impacts the performance of RL agents that explicitly exploit the Markovian assumption. For example, if we run q-learning and evaluate the performance of the greedy policy (i.e., without exploration) every $10,000$ steps, we see that q-learning does not converge. Instead, q-learning jumps between an optimal policy and a zero reward policy, as shown in Figure 3.

Now that we know that memory-augmented environments are not MDPs, we focus on proving that they are POMDPs. Such a proof can be found in Appendix A.1 and has important repercussions. In particular, all the theory for learning memoryless policies for POMDPs (Littman, 1994; Singh et al., 1994; Jaakkola et al., 1995; Li et al., 2011; Azizzadenesheli et al., 2018) also applies to memory-augmented environments. We explore this further in two parts: the prediction problem and the control problem. We then discuss the practical implications that follow from the theory.

## 5.1 HOW TO EVALUATE POLICIES IN MEMORY-AUGMENTED ENVIRONMENTS

For a given POMDP $\mathcal{P}$ and a memoryless policy $\pi(a|o)$, the policy prediction problem consists of estimating $q_\pi(o, a)$. Here, $q_\pi(o, a)$ is defined over observations and represents the expected discounted return when executing action $a$ given observation $o$ (at any time step $t$) and following $\pi$ thereafter: $q_\pi(o, a) = \mathbb{E}_\pi\left[\sum_{k=0}^{\infty} \gamma^k r_{t+k} \big| O_t = o, A_t = a\right]$. It is known that Monte-Carlo estimates are guaranteed to converge to the real values of $q_\pi(o, a)$, though they do have high variance. In contrast, TD estimates have lower variance but might not converge to $q_\pi(o, a)$ (Singh et al., 1994).

Failing to correctly estimate $q_\pi(o, a)$ is the reason behind q-learning's instability in the recall task (Figure 3). For instance, let $\pi$ be the optimal policy represented by blue arrows in OA1, then the real q-value for the red arrow $q_\pi(\emptyset, 2\top)$ is zero (the agent gets no reward if it executes $a_2$ in the first action). However, a one-step TD estimate would converge to $q_\pi(\emptyset, 2\top) = 0 + \gamma q_\pi(2, 3\top) = \gamma$. This is a problem since now $q_\pi(\emptyset, 2\top) > q_\pi(\emptyset, 1\top) = \gamma^2$ (for $\gamma \in (0, 1)$), and so q-learning will move from the current optimal policy $\pi$ to the zero reward policy that executes $2\top$ in $\emptyset$. We refer to these types of transitions as *non-Markovian shortcuts*. Note that, as Figure 3 shows, the B2 memory has more non-Markovian shortcuts than OA1. This is why q-learning over B2 is more unstable than q-learning over OA1 in this domain. More generally, we would expect that B$k$ memories introduce more non-Markovian shortcuts than OA$k$ memories since they are more flexible, which could partially explain the better empirical performance of OA$k$ and O$k$ memories.

There are two approaches that can mitigate this problem. The first is to use $n$-step TD estimates, with a large enough value of $n$. As Figure 4 shows, the performance of 20-step actor-critic in the gravity domain is far superior to 5-step actor-critic. The second is to increase the size of the memory, since doing so tends to remove non-Markovian shortcuts. This is also shown in Figure 4, as 5-step actor-critic performs better when using O2 or OA2, than when using O1 or OA1.

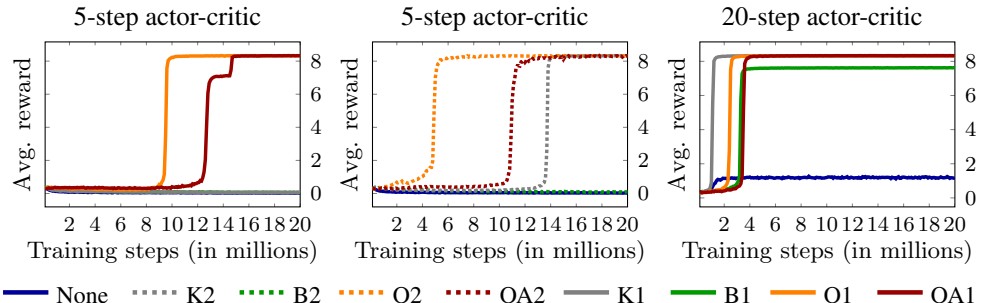

Figure 4: Experiments in the gravity domain. We reported the avg. reward per 100 steps.

## 5.2 HOW TO IMPROVE POLICIES IN MEMORY-AUGMENTED ENVIRONMENTS

We now focus our attention on the second part of the problem: how to use q-value estimates to find better policies. To do so, most RL algorithms exploit the policy improvement theorem. For MDPs, this theorem guarantees that updating the current policy $\pi$ by any amount towards the greedy policy $\tau(s) = \arg\max_{a \in A} q_\pi(s, a)$ will lead to better policies (Watkins, 1989; Sutton & Barto, 2018).

When learning memoryless policies for POMDPs, it is known that the policy improvement theorem only works locally (Jaakkola et al., 1995). To see why, note that the q-values over observations can be written in terms of q-values over states: $q_\pi(o, a) = \sum_{s \in S} P_\pi(s|o) q_\pi(s, a)$, for all $o \in O$, $a \in A$ (Singh et al., 1994). Here, $P_\pi(s|o)$ is the probability of being in state $s$ given that the observation is $o$ (at any time step), when following policy $\pi$. Intuitively, the policy improvement theorem does not work generally here because moving $\pi(o)$ towards $\tau(o) = \arg\max_{a \in A} q_\pi(o, a)$ increases the expectation over $q_\pi(s, a)$ without considering how $P_\pi(s|o)$ might change. Conversely, the policy improvement theorem works locally because updating $\pi$ by a small amount will also only have a small effect on $P_\pi(s|o)$, making such a difference insignificant. Therefore, a policy learning method that takes small update steps is guaranteed to converge to locally optimal memoryless policies—explaining why actor-critic converges smoothly in the gravity domain (Figure 2). Unfortunately, convergence to optimal memoryless policies is not guaranteed for general POMDPs.

Since memory-augmented environments are a form of POMDP, this local convergence guarantee also applies to them. As such, if the memory can represent the optimal policy, then that solution will be stable given accurate q-value estimations. This raises the question of what conditions for memory would guarantee convergence to a globally optimal memoryless policy. To investigate this topic, we considered an idealized version of Jaakkola et al. (1995)'s approach. This agent starts from a random policy $\pi$ and uses an oracle to compute $q_\pi(o, a)$ for all $o \in O$ and $a \in A$. Then, it moves $\pi(o)$ towards $\arg\max_{a \in A} q_\pi(o, a)$ a small step $\delta$ for all $o \in O$, and repeats.

Figure 5 shows the behaviour of this algorithm on a variant of the *recall task*. The rewards were selected to encourage convergence to suboptimal solutions (more details in Appendix B). In this environment, OA1 and B1 are enough to encode a memoryless policy that is globally optimal. However, note that OA1 converges to a suboptimal solution. Therefore, memory-augmented environments might converge to suboptimal solutions even if the memory is expressive enough to encode globally optimal policies. We do note that this problem vanishes as we increase the size of the memory in this domain. Unfortunately, convergence to an optimal memoryless policy cannot be guaranteed, even for memories that can model the belief states, as we prove for B$k$ in Appendix C.

## 5.3 SUMMARY: FROM THEORY TO PRACTICE

The theory suggests that the best approaches for learning effective memoryless policies in memory-augmented environments are methods that exploit the policy improvement theorem locally and evaluate policies using Monte-Carlo estimates (or n-step TD methods), such as n-step actor-critic, A3C (Mnih et al., 2016), or PPO (Schulman et al., 2017). Our empirical evidence also suggests the use of O$k$ or OA$k$ memories over B$k$ memories. While the core of our experimental analysis uses PPO, we also tested pure TD methods, including Sarsa($\lambda$) (Seijen & Sutton, 2014) and DDQN (Van Hasselt et al., 2016). Those results, which are shown in Appendix D.6, also favor O$k$ memories. Finally,

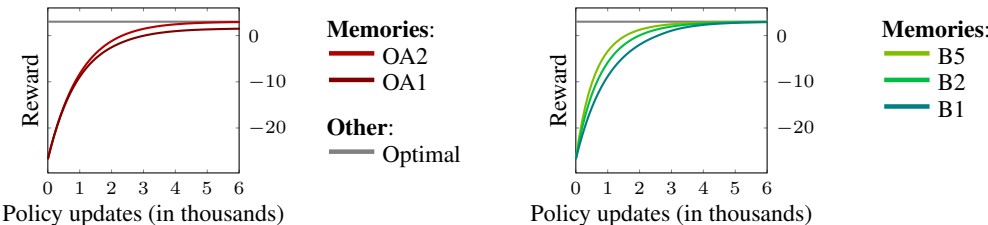

Figure 5: Tabular experiments in a variation of the recall task (details in Appendix B).

note that integrating external memories into existing RL toolkits is trivial. For instance, it takes less than 40 lines of code to integrate each external memory into OpenAI gym (Brockman et al., 2016).

## 6 EXPERIMENTAL EVALUATION

We ran experiments on a variety of environments with different types of external memory, including our new O$k$ and OA$k$ memories, as well as the existing k-order memories (K$k$) (Mnih et al., 2015) and binary memories (B$k$) (Littman, 1993). Below, we present results when using PPO (Schulman et al., 2017) and these memories. We also experimented with using no memory (None) and when using an LSTM. Figure 6 shows the results. Each line is the average reward per episode over 30 runs and the shadow area represents half a standard deviation. Details of the domains, hyperparameters, and network architectures can be found in Appendix D. We will release our code upon publication.

The left column shows results in the Hallway environments. These environments have been shown to be difficult for PPO with LSTM-based memory in previous work (Toro Icarte et al., 2019) and, indeed, we were also unable to get PPO with LSTMs to perform any better than a random policy. In contrast, PPO with OA6 and O6 memories is able to solve these tasks.

The middle column shows results in the MiniGrid environment (Chevalier-Boisvert & Willems, 2018). We experimented with the RedBlueDoors and MemoryS7 environments because they were specifically designed to test the agent's memory capabilities. We also decreased the agent's field of view from 8x8 to 3x3 cells to make these problems more challenging. In both cases, O3 and OA3 perform best, as they consistently converge to good solutions on all runs. In contrast, the LSTM performance was unreliable: we note that around half of the LSTM runs converged to poor policies.

The previous results used feed-forward networks for function approximation in grid-like domains. To test our approach in visually complex domains using convolutional networks, we also experimented with two Atari games: Pong and Seaquest. For these domains, we only gave the agent one frame of the game at a time (aside from the current memory state) and followed Machado et al. (2018)'s recommendations for making the environment stochastic. These domains are almost fully-observable, so it is unreasonable to expect O$k$, B$k$, or OA$k$ to outperform a k-order memory. Still, O3 has comparable performance to K3 in Pong and outperformed LSTMs in Seaquest. This shows that O$k$ memories can work well in visually complex domains. Note that OA3 performs well in Atari when trained by DDQN (see Appendix D.6) but it does not when using PPO.

Finally, we note that learning memoryless policies is usually faster than learning history-based policies. In fact, training PPO with an O$k$ memory was between $1.06$ to $9.85$ times faster than training PPO with an LSTM when using CPUs and between $1.71$ to $2.94$ times faster when using GPUs. The complete list of speedups can be found in Appendix D.5.

## 7 DISCUSSION, LIMITATIONS, AND FUTURE WORKS

Partial observability is a major challenge when applying RL in the real world. The most popular approaches for tackling partial observability are k-order memories and recurrent neural networks. These approaches are simple to implement, have strong empirical performance, and can be used off-the-shelf, without extra coding effort. The idea of allowing an agent to modify an external memory using primitive actions is similarly attractive since it can be easily implemented and combined with any RL method. Unfortunately, historically, such techniques have not worked well in practice

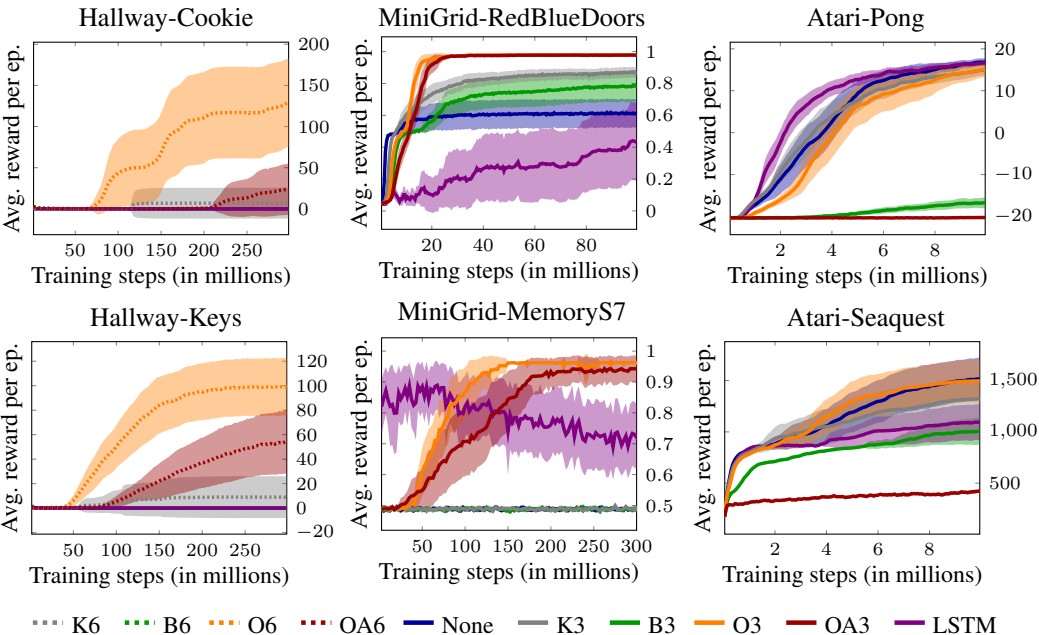

Figure 6: Results over partially observable benchmarks using PPO and different memories.

(Peshkin et al., 1999; Zhang et al., 2016). We revisited this idea and contributed a theoretical framework and new types of external memories. Notably, these memories outperformed k-order memories (Mnih et al., 2015), binary memories (Peshkin et al., 1999), and LSTM memories (Hausknecht & Stone, 2015) in most of our experiments – and were faster to train than LSTMs.

We view O$k$ and OA$k$ memories as the first step towards studying more expressive forms of memories. O$k$ and OA$k$ are limited by the size of their buffers and that the agent can only push observations into the buffer. This restricts the problems that can be solved. For instance, O$k$ memories cannot keep track of whether the current time step is odd or even (which a B1 memory could do). We believe that many of the limitations of O$k$ memories can be overcome by letting the agent determine a position in the buffer in which to save (or remove) an observation or alternatively by training an LSTM policy to control an O$k$ memory. It will also be interesting to study existing forms of memories in the context of memory-augmented environments. For instance, McCallum (1996) showed the effectiveness of tree-based memories in model-based RL and tape-like memories and stacks have worked well in supervised learning (Joulin & Mikolov, 2015; Zaremba & Sutskever, 2015).

Another avenue for future work is to further study the theory behind memory-augmented environments. We focused our analysis on the POMDP literature, but it is known that all theoretical guarantees given for function approximation in RL also apply to partial observability (Sutton & Barto, 2018, Chapter 17.3). Although the main conclusions that can be drawn from that body of work are similar to those described in Section 5.3, some recent works provide some interesting guarantees. For instance, the non-delusional q-learning algorithm, while impractical, is guaranteed to converge to globally optimal memoryless policies in memory-augmented environments (Lu et al., 2018).

## 8 CONCLUDING REMARKS

This work presented a lightweight approach to tackling partially observable RL. We provided the agent with an external memory and extra actions to write to it, and then used RL to learn a memoryless policy that jointly decides what to do and what to remember. This idea has been around since the 90s, but this is the first work to show how to make it work well in practice. The key step was to study the theory behind memory-augmented environments and to use that theory to propose novel forms of memories that support learning. Using the same RL agent, our external memories outperformed LSTM memories while being faster to train and trivial to implement. Our results suggests a broad array of topics for future exploration in the theory and practice of partially observable RL.

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

## A  FORMAL ANALYSIS

### A.1  MEMORY-AUGMENTED ENVIRONMENTS AS POMDPS

In this section, we show how to define memory-augmented environments as POMDPs. Given a memory-augmented environment $\mathcal{E} = \langle \mathcal{P}, \mathcal{M}_{\mathcal{P}} \rangle$, where $\mathcal{P} = \langle S, O, A, R, p, \omega, \gamma, \mu \rangle$ is a POMDP and $\mathcal{M}_{\mathcal{P}} = \langle M, W, \Gamma, \eta \rangle$ is an external memory module for $\mathcal{P}$, we define the POMDP $\mathcal{P}' = \langle S', O', A', R', p', \omega', \gamma', \mu' \rangle$ that corresponds to memory-augmented environment $\mathcal{E} = \langle \mathcal{P}, \mathcal{M}_{\mathcal{P}} \rangle$ as follows:

- $S' = S \times M \times O$
- $O' = M \times O$
- $A' = A \times W$
- $p'(\langle s', m', o' \rangle, r | \langle s, m, o \rangle, \langle a, w \rangle) = p(s', r | s, a) \Gamma(m' | m, w, o, a, r, o') \omega(o' | s')$
- $\omega'(\langle m', o' \rangle | \langle s, m, o \rangle) = \begin{cases} 1 & \text{if } m' = m \text{ and } o' = o \\ 0 & \text{otherwise} \end{cases}$
- $R' = R$
- $\mu'(s, m, o) = \mu(s) \eta(m) \omega(o | s)$

Note that $O$ has been included in $S'$ to ensure the consistency of $p'$ and $\omega'$ in the case that the next observation is stored in memory. Note also that, by construction, $\mathcal{P}'$ is equivalent to $\mathcal{E}$ because both environments generate rewards and observations following the same probability distributions from the agent's perspective.

## A.2 THE EXPRESSIVENESS OF AN EXTERNAL MEMORY MODULE

In this section, we show that if an external memory module is expressive enough, then the optimal memoryless policy for the memory-augmented environment corresponds to an optimal history-based solution to the original POMDP. We begin with the following definition:

**Definition A.1** (memory-update function). Given memory-augmented environment, $\mathcal{E} = \langle \mathcal{P}, \mathcal{M}_\mathcal{P} \rangle$, where POMDP $\mathcal{P} = \langle S, O, A, R, p, \omega, \gamma, \mu \rangle$ and external memory module $\mathcal{M}_\mathcal{P} = \langle M, W, \Gamma, \eta \rangle$, a *memory-update* function for $\mathcal{E}$ is defined as $f : M \times O \times A \to W$.

The memory-update function is akin to a deterministic policy for manipulating the memory (and only the memory). It does not determine how we select environment actions, it only dictates what memory-writing action to take once an environment action has been selected, given the current memory state, observation, and the action to be executed. Below, we define a criteria for an external memory module to be able to represent the optimal history-based policy for a POMDP, through the existence of a memory-updating function (i.e., a way for manipulating the memory) that can sufficiently summarize the interaction's history.

Let $\mathcal{H} = o_0, a_0, \ldots, a_{t-1}, o_t$ be the observation-action history for the original POMDP $\mathcal{P}$. We can use the memory-update function $f$ to construct a corresponding observation-action history $\mathcal{H}'$ in the memory-augmented environment, where each $o_i$ is replaced by a tuple $\langle m_i, o_i \rangle$ and each $a_i$ is replaced by a tuple $\langle a_i, w_i \rangle$. In particular, we can sample an initial memory state $m_0$ according to $\eta$, set the initial observation in the memory-augmented environment as $\langle m_0, o_0 \rangle$, set the first action as $\langle a_0, f(m_0, o_0, a_0) \rangle$, and set the next observation as $\langle o_1, m_1 \rangle$ where $m_1$ is sampled from $\Gamma$. This process can then be continued until the end of $\mathcal{H}$ to create a history $\mathcal{H}'$ for the memory-augmented environment.

In the analysis below, we will need to refer to the last memory state $m_t$ of such a history $\mathcal{H}'$. However, since $\eta$ and $\Gamma$ can be non-deterministic, there may be multiple valid memory-augmented histories $\mathcal{H}'$ that can be generated in this way for any $\mathcal{H}$. Thus, this generation process may result in any one of a set of last memory states. For any given history $\mathcal{H}$, we will refer to this set as $\Omega_f(\mathcal{H}) \in 2^M$.

Recall that the optimal policy for a POMDP $\mathcal{P} = \langle S, O, A, R, p, \omega, \gamma, \mu \rangle$ is a function of the history of an interaction: $\pi^*(a_t | o_0, a_0, \ldots, a_{t-1}, o_t)$. When the POMDP model is available, it can also be expressed in terms of *belief states*. A belief state at step $t$ is a probability distribution over (being in) each of the states in $S$, defined as $b_t : S \to [0, 1]$. The initial belief state is computed using the initial observation $o_0$: $b_0(s) \propto \omega(o_0 | s)$ for all $s \in S$. The belief state $b_{t+1}$ is then determined from the previous belief state $b_t$, the executed action $a_t$, and the resulting observation $o_{t+1}$ as $b_{t+1}(s') \propto \omega(o_{t+1} | s') \sum_{s \in S} p(s, a_t, s') b_t(s)$ for all $s' \in S$. In this way, the belief state correctly summarizes the history of an interaction, meaning that the optimal policy for $\mathcal{P}$ can then be written as a policy of the belief states $\pi^*(a_t | b_t)$. For convenience, we will let $b(\mathcal{H})$ be the belief state $b_t$ for history $\mathcal{H} = o_0, a_0, \ldots, a_{t-1}, o_t$.

We can now use the notion of a belief state to define the following:

**Definition A.2** (sufficiently expressive). Given a memory-augmented environment $\mathcal{E} = \langle \mathcal{P}, \mathcal{M}_\mathcal{P} \rangle$, an external memory module $\mathcal{M}_\mathcal{P}$ is *sufficiently expressive* for $\mathcal{P}$ if there exists a memory-update function $f$ for $\mathcal{E}$ such that for any two histories $\mathcal{H}^1 = o_0^1, a_0^1, \ldots, a_{t-1}^1, o_t^1$ and $\mathcal{H}^2 = o_0^2, a_0^2, \ldots, a_{i-1}^2, o_i^2$ where $o_t^1 = o_i^2$, the following holds:

$$\text{if } b(\mathcal{H}^1) \neq b(\mathcal{H}^2), \text{ then } \Omega_f(\mathcal{H}^1) \cap \Omega_f(\mathcal{H}^2) = \emptyset$$

Intuitively, given a memory augmented environment, $\mathcal{E} = \langle \mathcal{P}, \mathcal{M}_\mathcal{P} \rangle$ the external memory module $\mathcal{M}_\mathcal{P}$ is sufficiently expressive, if there is a way to use it to distinguish between belief states in the POMDP $\mathcal{P}$. If this holds, then there is a memoryless policy for the memory-augmented environment that can act differently (as needed) for each individual belief state. Thus, the following holds immediately:

**Proposition A.1.** *If an external memory module $\mathcal{M}_\mathcal{P}$ is sufficiently expressive for $\mathcal{P}$, then the optimal memoryless policy for the memory-augmented environment $\mathcal{E} = \langle \mathcal{P}, \mathcal{M}_\mathcal{P} \rangle$ is equivalent to the optimal history-based policy for $\mathcal{P}$.*

### A.3 Formal Definitions for K$k$ and OA$k$ Memories

In this section, we formally define k-order memories and OA$k$ memories as external memory modules. We begin with k-order memories:

**Definition A.3** (K$k$ memories). *For POMDP $\mathcal{P} = \langle S, O, A, R, p, \omega, \gamma, \mu \rangle$, a Kk external memory module for $\mathcal{P}$ of size $k$ is defined as $\mathcal{M}_\mathcal{P} = \langle M, W, \Gamma, \eta \rangle$, where $M = (O \cup \{\emptyset\})^k$, $W = \{\top\}$, $\eta(\emptyset^k) = 1$ (zero otherwise), and $\Gamma(m'|m, w, o, a, r, o') = 1$ if $m = \langle o^1, o^2, \cdots, o^k \rangle$ and $m' = \langle o^2, \cdots, o^k, o \rangle$ (zero otherwise).*

We now define OA$k$ memories as follows:

**Definition A.4** (OA$k$ memories). *For POMDP $\mathcal{P} = \langle S, O, A, R, p, \omega, \gamma, \mu \rangle$, an OAk memory for $\mathcal{P}$ is defined as $\mathcal{M}_\mathcal{P} = \langle M, W, \Gamma, \eta \rangle$, where $M = ((O \times A) \cup \{\emptyset\})^k$, $W = \{\top, \bot\}$, $\eta(\emptyset^k) = 1$ (zero otherwise), and $\Gamma(m'|m, w, o, a, r, o') = 1$ if $w = \bot$ and $m' = m$, or $w = \top$, $m = \langle e^1, e^2, \cdots, e^k \rangle$, and $m' = \langle e^2, \cdots, e^k, (o, a) \rangle$ where $e^i = (o^i, a^i)$ or $e^i = \emptyset$ (zero otherwise).*

### A.4 Sufficiency of K$k$, B$k$, O$k$, and OA$k$

In this section, we formally show that the external memory modules described in this work are sufficiently expressive if the buffer size is large enough:

**Proposition A.2.** *For any POMDP $\mathcal{P}$, the following holds:*

1. *Bk is sufficiently expressive for $\mathcal{P}$ if $k \geq \lceil \log_2 |O| \rceil + \lceil \log_2 |A| \rceil + \lceil \log_2 u \rceil$, where $u$ is the maximum number of possible belief states in the set of all histories that end in any given observation $o$. That is, $u$ is defined as*

$$u = \max_{o \in O} |\{b(\mathcal{H}) \mid \mathcal{H} \text{ is a history that ends in } o\}| \tag{1}$$

2. *OAk is sufficiently expressive for $\mathcal{P}$ if the belief state of any history is dependent on only the last $k$ observation-action pairs.*

3. *Ok and Kk are sufficiently expressive if the belief state of any history only depends on the last $k$ observations.*

*Proof.* For the case of B$k$, recall that to make a belief state update, we use the last belief state $b$, the last observation $o$, the last action $a$, and the current observation $o'$. Thus, we define the memory-update function to use the binary memory to store the first three of these components. Specifically, we can assign a unique integer to each of the possible observations, and use the first $\lceil \log_2 |O| \rceil$ bits to store the integer corresponding to $o$. We will do the same for the actions, and store the integer corresponding to the last action in the next $\lceil \log_2 |A| \rceil$ bits. Finally, we can uniquely assign an integer to each of the belief states that we could have been in when at $o$, and we will store that integer in the next $\lceil \log_2 u \rceil$ bits. Given the last belief state, last observation, and last action, the agent will therefore be able to distinguish between the belief state that they are in currently with observation $o'$. Thus, the memory-update function that correctly uses the bits in this way satisfies the conditions in definition A.2, and so B$k$ is sufficiently expressive for $\mathcal{P}$. Therefore, the statement holds for B$k$.

For the case of OA$k$, if the belief state only depends on the last $k$ observation-action pairs, then the memory-saving function that always saves will be able to distinguish between belief states. Thus OA$k$ is sufficiently expressive for $\mathcal{P}$ in this case. An analogous argument then applies for O$k$ and K$k$ in the conditions outlined above. $\qquad\square$

## B Variation of the Recall Task

We proposed a variation of the recall task to study whether OA$k$ memories can converge to suboptimal memoryless policies by locally improving its policy, as discussed in Section 5.2. This problem has only one observation $o$, two actions $A = \{0, 1\}$, and the discount factor is 1. The episode ends after executing 3 actions. The agent always receives no reward except when executing the last action. The final reward depends on the three actions executed during the episode according to the following table (where $a_i$ represents the $i$-th action):

| $(a_1, a_2)$ | R | $(a_1, a_2)$ | R |
|---|---|---|---|
| (0, 0) | $-5$ | (2, 0) | 0 |
| (0, 1) | 0.5 | (2, 1) | 0.5 |
| (0, 2) | 1 | (2, 2) | $-5$ |
| (0, 3) | 0.5 | (2, 3) | 0.5 |
| (1, 0) | 0 | (3, 0) | 0 |
| (1, 1) | 0.5 | (3, 1) | 0.5 |
| (1, 2) | $-0.5$ | (3, 2) | $-5$ |
| (1, 3) | 0.75 | (3, 3) | 0.5 |

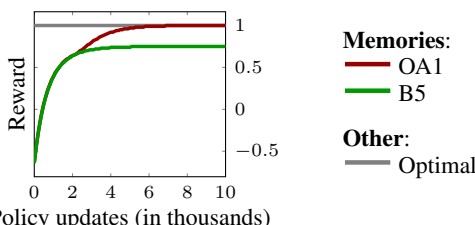

Figure 7: Variation of the recall task where B$k$ always converges to a suboptimal policy.

| $(a_1, a_2, a_3)$ | R | $(a_1, a_2, a_3)$ | R |
|---|---|---|---|
| (0, 0, 0) | 0 | (1, 0, 0) | -100 |
| (0, 0, 1) | 2 | (1, 0, 1) | -100 |
| (0, 1, 0) | 3 | (1, 1, 0) | -10 |
| (0, 1, 1) | 1 | (1, 1, 1) | -10 |

While this problem can be optimally solved using an OA1 memory, the ideal agent from Section 5.2 converges to a suboptimal policy. The reason is that an optimal policy would execute action $\langle 0, \top \rangle$ when the memory buffer is $\emptyset$, action $\langle 1, \top \rangle$ when the buffer contains $\langle o, 0 \rangle$, and action $\langle 0, \top \rangle$ when the buffer contains $\langle o, 1 \rangle$. However, if the agent, while exploring, executes action $\langle 1, \top \rangle$ in the first action, then executing $\langle 0, \top \rangle$ from $\langle o, 1 \rangle$ will give a expected return of -100. As this large penalty will be considered when estimating $q_\pi(\langle o, 1 \rangle, \langle 0, \top \rangle)$, the agent will prefer action $\langle 1, \top \rangle$ over the optimal action $\langle 0, \top \rangle$ in $\langle o, 1 \rangle$ – causing the convergence to a suboptimal policy.

## C  CONVERGENCE TO SUBOPTIMAL SOLUTIONS

In this section, we show that having a sufficiently expressive external memory module is not enough to guarantee that a policy-improvement based algorithm will find a memoryless policy that is optimal for the original POMDP. We do so by showing that the Jaakkola et al. (1995) algorithm will get stuck in a local minimum when using B$k$ memory, for any value of $k$, on a variant of the recall task described in Section 5. We refer to this problem as the 4-*action recall task*. In this case, an episode consists of 2 steps, and there are four possible actions that can be taken. We denote these actions as $\mathcal{A} = \{0, 1, 2, 3\}$. The rewards received at the end of the episode is shown in the table in Figure 7. That figure also shows (empirically) that a B5 memory converges to a suboptimal policy in this problem. We also note that the optimal policy is given by selecting action 0 and then action 2. If any other action is taken as the first action, then the optimal action is to take action 3.

We now show that B2 is sufficiently expressive for the 4-action recall task, by identifying a policy for the task. The memory state will start with a value 00. When the memory is 00, action 0 should be taken along with the write action 01. Then action 2 should be taken for maximal reward. The memory write action can be arbitrary at this point since the episode is over. If, however, any action other than 0 is taken in the initial state when the memory is 00, then the write action should be 11. In the resulting state, action 3 will be taken along with any arbitrary memory writing action.

We will see that the q-values for the policy that always takes $(1, 3)$ is a local optimum to this problem. This is shown for B1 in Lemma C.2, and generalized later. As a result, algorithms that follow a policy improvement scheme can get stuck at this optimum. We show that this can happen when using an idealized version of the Jaakkola et al. (1995) algorithm, even when starting from a uniform policy over all actions. This algorithm alternates between policy evaluation and policy improvement. During policy improvement, the current policy $\pi$ is moved a step closer to the policy $\pi^g$ that is greedy on the current q-values of $\pi$. That is, for any observation $o$ and action $a$, the new policy $\pi'$ is given by $\pi'(a \mid o) = (1 - \varepsilon)\pi(a \mid o) + \varepsilon \pi^g(a \mid o)$, where $\varepsilon$ is an input parameter where $0 \leq \varepsilon \leq 1$. For our analysis, we assume that before every policy improvement update, we computed the actual q-values of the current policy. We refer to this algorithm as *the Jaakkola et al. (1995) algorithm with perfect q-value estimates*. For simplicity, we assume the algorithm breaks ties between write actions in favour of the higher valued binary number (*i.e.* 1 over 0 or 11 over 10). Under this scenario, we will

see that a sufficiently expressive external memory module is not enough to guarantee convergence to a policy that is as good as the optimal history-based policy. This is shown in Theorem C.6

We first prove that this algorithm fails to converge to the optimal policy from the starting state in the case of B1, and then show how the argument can be extended to any B$k$. Let us now introduce some notation for the case of $k = 1$. We will use $p_{aw}^m$ as short for $\pi(\langle a, w \rangle \mid \langle o, m \rangle)$ and $Q_{aw}^m$ as short for $\mathbb{E}_\pi[R_0 + \gamma R_1 + ... \mid m, a, o, w]$. Note that we omit the environment observation $o$ since there is only one observation in this problem and use a discount factor of 1. We also define $p_a^m$ as the probability given memory state $m$ of action $a$ marginalized over memory-write actions (i.e. $p_{a0}^m + p_{a1}^m$).

**Lemma C.1.** Let $b_i = \frac{p_{i0}^0}{1+\sum_j p_{j0}^0}$ and $t_i = \frac{p_{i1}^0}{\sum_j p_{j1}^0}$. The q-values for a given policy $\pi$ for the 4-action recall task with B1 memory are given by:

$$Q_{aw}^0 = \sum_{0 \le i \le 3} r_{ia} b_i + \left(1 - \sum_{0 \le i \le 3} b_i\right) \sum_{0 \le i \le 3} p_i^w r_{ai}$$

$$Q_{aw}^1 = \sum_{0 \le i \le 3} t_i r_{ia}$$

*Proof.* We consider the cases of $m = 0$ and $m = 1$ separately. Let $s^*$ denote the initial state of the POMDP, and $s_0, s_1, s_2, s_3$ as the states reached after performing actions $0, 1, 2, 3$ respectively, in the initial state.

Case $m = 0$:

$$Q_{aw}^0 = \sum_{0 \le i \le 3} r_{ia} \mathbb{P}_\pi[S = s_i | M = 0] + \mathbb{P}_\pi[S = s^* | M = 0] \mathbb{E}_\pi[R_0 + \gamma R_1 + ... | M = w, S = s_a]$$

$$= \sum_{0 \le i \le 3} r_{ia} \mathbb{P}_\pi[S = s_i | M = 0] + \mathbb{P}_\pi[S = s^* | M = 0] \sum_{0 \le i \le 3} p_i^w r_{ai}$$

Now, let us turn to $\mathbb{P}_\pi[S = s_i | M = 0]$:

$$\mathbb{P}_\pi[S = s^* | M = 0] = 1 - \sum_{0 \le i \le 3} \mathbb{P}_\pi[S = s_i | M = 0]$$

For each $\mathbb{P}_\pi[S = s_i | M = 0]$, we now get the following:

$$\mathbb{P}_\pi[S = s_i | M = 0] = \frac{\mathbb{P}_\pi[S = s_i, M = 0]}{\mathbb{P}_\pi[M = 0]} = \frac{\frac{p_{i0}^0}{2}}{\frac{1}{2} + \frac{1}{2}\sum_{j \in \{0,1,2,3\}} p_{j0}^0} = b_i$$

We note that the $1/2$ factors come from the fact that if we ran an infinite number of episodes of this task, half the encountered states would be $s^*$ (since every episode starts there), and half would be the result of taking an action in $s^*$.

By substituting this into the expression above, we see the statement holds in this case.

Case $m = 1$:

$$\mathbb{E}_\pi[R_0 + \gamma R_1 + ... | M = 1, A = a, W = w] = \sum_{0 \le i \le 3} r_{ia} \mathbb{P}_\pi[S = s_i | M = 1]$$

Again, when we compute $\mathbb{P}_\pi[S = s_i | M = 1]$, we get the following:

$$\mathbb{P}_\pi[S = s_i | M = 1] = \frac{\mathbb{P}_\pi[M = 1, S = s_i]}{\mathbb{P}_\pi[M = 1]} = \frac{\frac{1}{2}p_{i1}^0}{\frac{1}{2}\sum_{0 \le j \le 3} p_{j1}^0} = t_i$$

The $1/2$ factors emerge for the same reason as discussed above. Again, with substitution, this recovers the desired q-value expression. □

Let us now show that the suboptimal policy that selects action 1 when the memory state is 0 and action 3 when the memory state is 1 is a local optimum:

**Lemma C.2.** *Let $\pi_l$ be the policy for the 4-action recall task with B1 memory with $p^0_{11} = 1$ and $p^1_{31} = 1$. Then the policy that is greedy on the q-values of $\pi_l$ is $\pi_1$ itself.*

*Proof.* Consider the q-values of $\pi_l$ given in Lemma C.1. Since $b_i = 0$, $Q^0_{aw}$ simplifies to $\sum_{0 \leq i \leq 3} p^w_i r_{ai}$. Since $p^1_3 = 1$ and $p^1_i = 0$ for $i \neq 3$, this means that $Q^0_{11} = 1 \cdot r_{13} = 3/4$, and $Q^0_{a1} = 1 \cdot r_{a3} = 1/2$. Since $p^0_1 = 1$ and $p^0_i = 0$ for $i \neq 1$, we also have that $Q^0_{a0} = 1/2$ for all $a$.

Now notice that $t_1 = 1$, and $t_i = 0$ for $i \neq 1$. As such $Q^1_{aw}$ simplifies to $r_{1a}$.

Let us now consider the greedy policy $\pi^g$ on these q-values. In the case that the memory state is 0, the greedy action over $Q^0_{11} = 3/4$ and $Q^0_{aw} = 1/2$ for any $a$ and $w$ where $a \neq 1$ or $w \neq 1$ is clearly to set $p^0_{11} = 1$. In the case that the memory state is 1, the greedy action is the one that maximizes $r_{1a}$ which is $a = 3$ as that gives $3/4$. Therefore, $\pi^g$ will set $p^1_{31} = 1$ since it tiebreaks in favour of a higher memory write actions. Since $\pi^g$ is clearly the same policy as $\pi_l$, the statement is true. $\quad\square$

We will now show that the Jaakkola et al. (1995) algorithm with perfect q-value estimation will converge to this local optimum. We begin with the following lemma, which will used for the inductive step in the full result.

**Lemma C.3.** *Let $\varepsilon$ be a constant such that $0 \leq \varepsilon \leq 1$. Suppose that $\pi$ is a policy for the 4-action recall task with B1 memory, such that the policy that is greedy on the q-values of $\pi$ is $\pi_l$. Let $\tilde{\pi}$ be defined such that for any $\tilde{\pi} = (1 - \varepsilon)\pi + \varepsilon\pi_l$. Then the policy that is greedy on the q-values of $\tilde{\pi}$ is also $\pi_l$.*

*Proof.* In this proof, we will use $\tilde{Q}^m_{aw}$ for the q-values for $\tilde{\pi}$. We will similarly define $\tilde{p}^m_{aw}$, $\tilde{b}_i$, and $\tilde{t}_i$ as $p^m_{aw}$, $b_i$, and $t_i$ were defined above. Notice that $\tilde{p}^0_{11} = (1 - \varepsilon)p^0_{11} + \varepsilon$, $\tilde{p}^1_{31} = (1 - \varepsilon)p^1_{31} + \varepsilon$, and $\tilde{p}^m_{aw} = (1 - \varepsilon)p^m_{aw}$, otherwise. We will also express $\tilde{p}^w_{11} = (1 - \varepsilon)p^w_{11} + \varepsilon\mathbb{I}_{w=0}$ where $\mathbb{I}$ is the indicator function. We now consider the q-values in the two cases of $m = 0$ and $m = 1$.

Case $m = 0$:

We begin by expressing $\tilde{b}_i$ in terms of $b_i$.

$$\tilde{b}_i = \frac{\tilde{p}^0_{i0}}{1 + \sum_j \tilde{p}^0_{j0}} = \frac{(1 - \varepsilon)p^0_{i0}}{1 + (1 - \varepsilon)\sum_j p^0_{j0}} \frac{1 + \sum_j p^0_{j0}}{1 + \sum_j p^0_{j0}} = \frac{(1 - \varepsilon)(1 + \sum_j p^0_{j0})}{1 + (1 - \varepsilon)\sum_j p^0_{j0}} b_i = (1 - \varepsilon)\alpha b_i \tag{2}$$

where $\alpha = (1 + \sum_j p^0_{j0})/(1 + (1 - \varepsilon)\sum_j p^0_{j0})$. In addition, we have the following expression:

$$1 - \sum_{0 \leq i \leq 3} b_i = \frac{1 + \sum_{0 \leq j \leq 3} p^0_{j0}}{1 + \sum_{0 \leq j \leq 3} p^0_{j0}} - \sum_{0 \leq i \leq 3} \frac{p^0_{i0}}{1 + \sum_{0 \leq j \leq 3} p^0_{j0}} = \frac{1}{1 + \sum_{0 \leq j \leq 3} p^0_{j0}} \tag{3}$$

Clearly, a similar expression exists for $1 - \sum_i \tilde{b}_i$. We will now use these expressions in the following derivation, which begins with the q-vale expression from Lemma C.1:

$$\tilde{Q}^0_{aw} = \sum_{0 \leq i \leq 3} r_{ia}\tilde{b}_i + \left(1 - \sum_{0 \leq i \leq 3} \tilde{b}_i\right) \sum_{0 \leq i \leq 3} \tilde{p}^w_i r_{ai}$$

$$= \sum_{0 \leq i \leq 3} r_{ia}\tilde{b}_i + \left(\frac{1}{1 + \sum_j \tilde{p}^0_{j0}}\right) \sum_{0 \leq i \leq 3} \tilde{p}^w_i r_{ai}$$

$$= \sum_{0 \leq i \leq 3} r_{ia}\tilde{b}_i + \left(\frac{1}{1 + (1 - \varepsilon)\sum_j p^0_{j0}}\right) \sum_{0 \leq i \leq 3} \tilde{p}^w_i r_{ai}$$

$$= \sum_{0 \le i \le 3} r_{ia} \tilde{b}_i + \alpha \left( \frac{1}{1 + \sum_j p_{j0}^0} \right) \sum_{0 \le i \le 3} \tilde{p}_i^w r_{ai}$$

$$= \sum_{0 \le i \le 3} r_{ia}(1 - \varepsilon)\alpha b_i + \alpha(1 - \sum_{0 \le i \le 3} b_i) \sum_{0 \le i \le 3} \tilde{p}_i^w r_{ai}$$

Recall that $\tilde{p}_i^w = (\tilde{p}_{i0}^w + \tilde{p}_{i1}^w)$. Thus, we can continue the derivation as follows:

$$\tilde{Q}_{aw}^0 = \sum_{0 \le i \le 3} r_{ia}(1 - \varepsilon)\alpha b_i + \alpha(1 - \sum_{0 \le i \le 3} b_i) \sum_{0 \le i \le 3} (\tilde{p}_{i0}^w + \tilde{p}_{i1}^w) r_{ai}$$

$$= (1 - \varepsilon)\alpha \sum_{0 \le i \le 3} r_{ia} b_i$$

$$+ \alpha(1 - \sum_{0 \le i \le 3} b_i) \left( \varepsilon \mathbb{I}_{w=0} r_{a1} + \varepsilon \mathbb{I}_{w=1} r_{a3} + (1 - \varepsilon) \sum_{0 \le i \le 3} (\tilde{p}_{i0}^w + \tilde{p}_{i1}^w) r_{ai} \right)$$

$$= (1 - \varepsilon)\alpha \left( \sum_{0 \le i \le 3} r_{ia} b_i + (1 - \sum_{0 \le i \le 3} b_i) \sum_{0 \le i \le 3} p_i^w r_{ai} \right) +$$

$$+ \varepsilon\alpha(1 - \sum_{0 \le i \le 3} b_i) \left( \mathbb{I}_{w=0} r_{a1} + \mathbb{I}_{w=1} r_{a3} \right)$$

$$= \alpha \left[ (1 - \varepsilon) Q_{aw}^0 + \varepsilon(1 - \sum_{0 \le i \le 3} b_i) \left( \mathbb{I}_{w=0} r_{a1} + \mathbb{I}_{w=1} r_{a3} \right) \right]$$

Notice that the above is merely a linear combination (with non-negative coefficients) of $Q_{aw}^0$ and $\mathbb{I}_{w=0} r_{a1} + \mathbb{I}_{w=1} r_{a3}$. The remaining scalars do not depend on $a$ or $w$. Now $Q_{aw}^0$ has its maximum when $a = 1$ and $w = 1$ because this was the greedy action over the q-values of $\pi$. We also see that $\mathbb{I}_{w=0} r_{a1} + \mathbb{I}_{w=1} r_{a3}$ has its maximum when $a = 1$ and $w = 1$ by the way the reward function is set up. Thus, the greedy policy over the q-values of $\tilde{\pi}$ in memory state 0, takes $a = w = 1$, which is the same as $\pi_l$.

Case $m = 1$:

By Lemma C.1, we have the following:

$$\tilde{Q}_{aw}^1 = \sum_{0 \le i \le 3} \frac{\tilde{p}_{i1}^0}{\sum_j \tilde{p}_{j1}^0} r_{ia}$$

$$= \frac{(1 - \varepsilon) \left( \sum_{0 \le i \le 3} p_{i1}^0 r_{ia} \right) + \varepsilon r_{1a}}{(1 - \varepsilon) \sum_j p_{j1}^0 + \varepsilon}$$

Recall that $Q_{aw}^1$ had the maximal value for $a = 3$ and $w = 1$, since these correspond to the greedy action in $\pi_l$. Since the denominator in $t_i$ is a constant, this means that the numerator, $\sum_{0 \le i \le 3} p_{i1}^0 r_{ia}$, takes on its maximum value when $a = 3$ and $w = 1$. Further, $r_{1a}$ is maximized for $a = 3$. Thus, the numerator in the expression above is maximized when $a = 3$ and $w = 1$. Since the denominator is a constant, $\tilde{Q}_{aw}^1$ is also maximized with these values. So the greedy policy on $\tilde{\pi}$, when $m = 1$, also agrees with $\pi_l$.

Since these are all the possible memory states in B1, the greedy policy for $\tilde{\pi}$ must also be $\pi_l$.  $\square$

We can now use this lemma as the inductive step to prove that the Jaakkola et al. (1995) algorithm with perfect q-value estimates will converge to this suboptimal policy on the 4-action recall task when using B1 memory.

**Lemma C.4.** *The Jaakkola et al. (1995) algorithm with perfect q-value estimates will converge to a suboptimal policy on the 4-action recall task with B1 external memory, when initialized with a uniform policy.*

*Proof.* Let $\varepsilon$ be the fixed value that the policy is moved towards the greedy policy on every step, where $0 \leq \varepsilon \leq 1$. The proof is by induction. We will show that on every step of the algorithm, $\pi_l$ will be the policy that is greedy on the current policy. Thus, on every step, the current policy will get closer to $\pi_l$.

Base Case: We begin at the uniform policy $\pi_u$. Putting the values in to the expressions in Lemma C.1, we see that the q-values for this policy are

| $(a, w)$ | $m = 0$ | $m = 1$ |
|---|---|---|
| $(0, 0)$ | $-0.916$ | $-1.25$ |
| $(0, 1)$ | $-0.916$ | $-1.25$ |
| $(1, 0)$ | $0.2916$ | $0.5$ |
| $(1, 1)$ | $0.2916$ | $0.5$ |
| $(2, 0)$ | $-1.4583$ | $-2.375$ |
| $(2, 1)$ | $-1.4583$ | $-2.375$ |
| $(3, 0)$ | $-0.47916$ | $0.5625$ |
| $(3, 1)$ | $-0.47916$ | $0.5625$ |

Given the tiebreaking rules defined, we see that the greedy actions are the same as those taken by $\pi_l$.

Inductive Step: Assume that the policy that is greedy on the q-values of the current policy $\pi$ is equivalent to $\pi_l$. Thus, the next policy is $\tilde{\pi} = (1 - \varepsilon)\pi + \epsilon\pi_l$. By Lemma C.3, the policy that is greedy on the q-values of $\tilde{\pi}$ is also $\pi_l$.

Thus, from initialization of $\pi_u$, we see that the on every policy improvement step, the policy will get closer to $\pi_l$, which it much reach in the limit. Since this policy is not as good as the best history-based policy, the statement holds. □

One may be tempted to think that adding more memory would solve this problem, i.e., it will allow the agent to learn a policy that is as good as the optimal history-based policy for mePOMDP $\mathcal{P}$, since B$k$ is sufficiently expressive for $k \geq 2$. However, the policy improvement mechanism cannot guarantee that this is learned. We see this below.

**Lemma C.5.** *The Jaakkola et al. (1995) algorithm with perfect q-value estimates will converge to a suboptimal policy on the 4-action recall task with B$k$ external memory, when initialized with a uniform policy, for any $k \geq 1$.*

*Proof.* We generalize the argument in Lemmas C.3 and C.4 to B$k$ memory. This gives $2^k$ possible memory states $0, \ldots, 2^k - 1$. We note that in this case, due to tiebreak rules, the locally optimal, globally suboptimal policy $\pi_l$ is given by $p^0_{1(2^k-1)} = p^m_{3(2^k-1)} = 1$ (where $m \geq 1$), an $p^m_{aw} = 0$ otherwise. Notice that analogous formulas as given in Lemma C.1 for the q-values can be constructed for an arbitrary $k$. In the case of $Q^0_{aw}$, the expression stays as is, there are just more options for $w$. For convenience, we will refer to $b_i$ from Lemma C.1 as $b^k_i$.

In the case of $Q^n_{aw}$ for $n \geq 1$ (the analogue of $Q^1_{aw}$ in Lemma C.1), the only change is that $t_i$ is replaced by a $t^n_i = \frac{p^0_{im}}{\sum_i p^0_{im}}$. We note that the reason this has stayed so similar is because there are only two points of decision-making in this task, and the memory state can only be non-zero in the second one since it is always initialized to 0.

Now we will prove a similar statement as Lemma C.4: when the Jaakkola et al. (1995) algorithm is initialized to the uniform policy, the policy improvement step will always move the policy closer to $\pi_l$, since $\pi_l$ is the policy that is greedy on the current q-values. The proof is by induction.

Base Case: We begin with the uniform policy. We will proceed by identifying the relation between various quantities in the case of $k = 1$ and for an arbitrary $k > 1$. This will allow us to re-use the computation performed in the base case of Lemma C.4.

Notice that because there are always 4 environment actions applicable and $2^k$ memory write actions, there are $2^{k+2}$ possible actions in every state. Thus $p_{aw}^m = 1/2^{k+2}$. Let $q_{aw}^m = 1/2^3$ be this value in the case that $k = 1$. Clearly, $p_{aw}^m = \frac{q_{aw}^m}{2^{k-1}}$.

Now, let us compare the value of $b_i^k$ relative to $b_i^1$ (which was $b_i$ in Lemma C.1 for the case of $k = 1$):

$$b_i^k = \frac{p_{i0}^0}{1 + \sum_j p_{j0}^0} = \frac{\frac{1}{2^{k-1}} q_{i0}^0}{1 + \frac{1}{2^{k-1}} \sum_j q_{j0}^0} = \frac{1}{2^{k-1}} \times \frac{1 + \sum_{0 \le j \le 3} q_{j0}^0}{1 + \frac{1}{2^{k-1}} \sum_{0 \le j \le 3} q_{j0}^0} b_i^1 = \frac{\beta^k}{2^{k-1}} b_i^1$$

where $\beta^k = (1 + \sum_{0 \le j \le 3} p_{j0}^0)/(1 + \frac{1}{2^{k-1}} \sum_{0 \le j \le 3} p_{j0}^0)$.

We will now use this expression, and the fact that the marginalized probabilities $p_a^m = q_a^m = 1/4$, to do the following derivation:

$$Q_{aw}^0 = \sum_{0 \le i \le 3} r_{ia} b_i^k + \left(1 - \sum_{0 \le i \le 3} b_i^k\right) \sum_{0 \le i \le 3} p_i^w r_{ai} \tag{4}$$

$$= \frac{\beta^k}{2^{k-1}} \sum_{0 \le i \le 3} r_{ia} b_i^1 + \beta^k \left(1 - \sum_{0 \le i \le 3} b_i^1\right) \sum_{0 \le i \le 3} q_i^w r_{ai} \tag{5}$$

$$= \frac{\beta^k}{2^{k-1}} \left[\sum_{0 \le i \le 3} r_{ia} b_i^1 + \left(1 - \sum_{0 \le i \le 3} b_i^1\right) \sum_{0 \le i \le 3} q_i^w r_{ai}\right] \tag{6}$$

$$+ \left(\frac{2^{k-1} - 1}{2^{k-1}}\right) \beta^k \left(1 - \sum_{0 \le i \le 3} b_i^1\right) \sum_{0 \le i \le 3} q_i^w r_{ai} \tag{7}$$

We note that the second line follows by a similar identity to one used in expression 3.

The first term in this last expression (*ie.* on line 6) is just the same as the q-value in the case that $k = 1$ is multiplied by a constant. It is therefore maximized by the values $a = 1$ and $w = 1$ as shown in the base case of Lemma C.2. The second term (*ie.* line 7) is maximized for the average rewards seen after any first action (since the uniform policy is in use). By inspection, we can see this happens for $a = 1$ and it is equal for the different memory writing actions. Thus, by our tie-breaking definition, the greedy action will be $a = 1$ and $w = 2^k - 1$ and so the base case holds when the memory state is 0.

Let us now consider $Q_{aw}^m$ for $m \ge 1$. Here, the $t_i^n$ are exactly the same as the value of $t_i$ in the case that $k = 1$, when the uniform policy is in use. So applying the same tie-breaking rules means that the greedy action is $a = 3, w = 2^k - 1$, as predicted.

Inductive step: We now consider how B$k$ for $k > 1$ is updated. For $m = 0$, we can use the same argument as used in Lemma C.3, simply replacing $\epsilon \mathbb{I}_{w=1} r_{a3}$ with $\epsilon \sum_{i=1}^{2^k - 1} \mathbb{I}_{w=i} r_{a3}$. For $m \ge 1$, we have that $Q_{aw}^m$ is updated in the exact same way for $m = 2^k - 1$ as in the proof in Lemma C.3 for $m = 1$, and further that $1 \le m < 2^k - 1$ have $Q$-values which remain unchanged, as the conditional probability $t_a^m$ of having taken action $a$ on having written $m$ does not change since $p_{am}^0$ are all simply scaled by $1 - \varepsilon$ for these values of $m$. As such, the greedy actions will again be those given by $\pi_l$.

Thus, every step of policy improvement during the Jaakkola et al. (1995) algorithm with perfect q-value estimates with move closer to $\pi_l$ when starting from the uniform policy. Since $\pi_l$ is suboptimal, the statement is true. □

We can now prove our main result: that the Jaakkola et al. (1995) algorithm is not guaranteed to converge to a policy that is as good as best history-based policy, even when using sufficiently expressive memory and perfect q-value estimations. This follows from Lemma C.5 and the fact that B2 is sufficiently expressive for the recall task used in this section.

**Theorem C.6.** *There exists a POMDP $\mathcal{P}$, such that even with a sufficiently expressive external memory module $\mathcal{M}$, the Jaakkola et al. (1995) algorithm with perfect q-value estimates will not*

*converge to the optimal memoryless policy from an initially uniform policy $\pi_u$, when used on the memory-augmented environment $\langle \mathcal{P}, \mathcal{M} \rangle$.*

Further, another result can be seen from this very same domain, by modifying $\mathcal{P}$.

**Proposition C.1.** *There exists a POMDP $\mathcal{P}$, such that even with a sufficiently expressive external memory module $\mathcal{M}$, the Jaakkola et al. (1995) algorithm with perfect q-value estimates will converge to a policy that is arbitrarily far from the optimal memoryless policy from an initially uniform policy $\pi_u$, when used on the memory-augmented environment $\langle \mathcal{P}, \mathcal{M} \rangle$*

*Proof.* This proof uses the same technique given by Singh et al. (1994). To do so, we merely have to scale the rewards used in the 4-action recall task by any constant $a \geq 1$. Since the ordering of the action values remains the same, the same arguments will apply. Thus the algorithm will converge to a policy that receives $\frac{a}{4}$ less reward than the optimal history-based policy, which can be made arbitrarily large by choice of $a$. □

The above counterexample $\mathcal{P}$ may further give some indication as to why OA-type memory may be empirically outperforming B-type memory in our experiments. A large part of the reason the above counterexample fails to hold is because the states after the first action become confounded, as the agent does not learn to discern, even with arbitrarily large memory, between the different actions to trigger the storage of a given memory state.

For empirical confirmation of this, see Figure 7 showing B5 converging to $\pi_l$ and OA1 converging to $\pi^*$ in POMDP $\mathcal{P}$. Intuitively, more pre-packaged knowledge in memory, and more structure given to the agent via the memory write/read mechanism, means the memory-augmented agent has fewer ways that it can confound itself in learning how to use memory, which is why choosing memory that comes with more built-in structure helps avoid local minima which are caused by this confusion.

## D    EXPERIMENTAL EVALUATION

This section provides further details about our experimental section.

### D.1    TABULAR EXPERIMENTS IN THE GRAVITY DOMAIN AND RECALL TASK

In the experiments with tabular q-learning, all the memories were tested using the same hyperparameters: They explored using $\epsilon$-greedy with $\epsilon = 0.01$, the discount factor was $0.95$, and the learning rate was $0.1$. We also used an optimistic initialization for the q-values.

In the experiment with n-step actor-critic, all the memories were also tested using the same hyperparameters. The discount factor was $0.95$. We used a soft-max exploratory policy initialized to a uniform distribution. The learning rate for the policy was $0.1$ and the learning rate for the value function was $0.001$.

### D.2    THE HALLWAY DOMAINS

Figure 8 shows an overview of the Hallway environments (Toro Icarte et al., 2019). These environments consist of three rooms connected by a hallway. The agent (shown as a purple triangle) can move in the four cardinal directions but its actions fail with a 5% probability. These are partially observable environments since the agent can only see what it is in the room that it currently occupies, as shown in Figure 8a. What makes these tasks difficult is the hallway. The hallway forces the agent to observe long sequences of identical observations (multiple times) to solve a task. However, depending on previous observations, the optimal actions (and expected rewards) will be completely different when the agent is in the hallway.

The cookie domain is shown in Figure 8b. In this domain, there is a button in the yellow room that, when pressed, causes a cookie to randomly appear in the red or blue room. The agent receives a reward of $+1$ for eating the cookie and may then go and press the button again. Pressing the button before eating the cookie removes the existing cookie and delivers a new cookie. Each episode is $10,000$ steps long, during which the agent should attempt to eat as many cookies as possible.

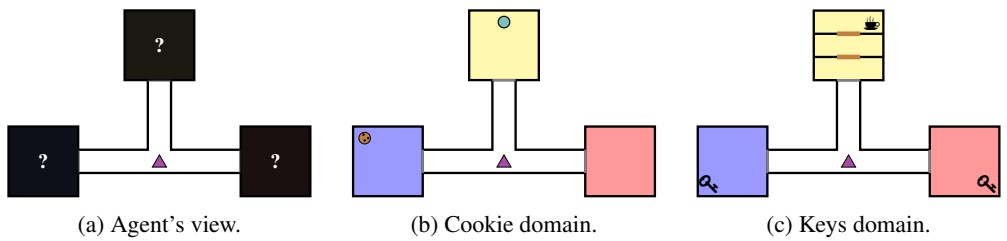

(a) Agent's view.    (b) Cookie domain.    (c) Keys domain.

Figure 8: The hallway environments.

Figure 8c shows another hallway environment, called the keys domain. Here, the agent receives a reward of +1 when it reaches the coffee. To do so, it must open the two doors (shown in brown). Each door requires a different key to open it, and the agent can only carry one key at a time. Initially, the two keys are randomly located in either the blue room, the red room, or split between them. When the coffee is reached, the agent is relocated in the middle of the hallway, the doors are locked, and the two keys are randomly placed in the blue and red rooms. Each episode is 10,000 steps long, during which the agent should attempt to reach the coffee as many times as possible.

While interacting with the environment, the agent is given a "top-down" view of the world represented as a set of binary matrices. The first matrix has a 1 in the current location of the agent, the second has a 1 in only those locations that are currently observable, and the remaining matrices each correspond to an object in the environment. These remaining matrices only have a 1 at those locations that were both currently observable and contained that object (i.e., locations in other rooms are "blacked out").

We ran experiments using the OpenAI baseline implementation of PPO (Hesse et al., 2017). For O6, OA6, K6, and B6, the neural network used had 4 fully connected layers with 512 neurons per layer with *tanh* activation functions. The LSTM baseline used the same network with an extra LSTM layer of 512 neurons. All approaches were trained using the Adam optimizer (Kingma & Ba, 2014).

The approaches O6, OA6, K6, and B6 used a learning rate of 1e-5, a clipping range of 0.1, 16 training minibatches per update, and one training epoch per update. The value of n used for the n-step TD updates was 2048. The rest of the hyperparameters were set to their default values (Hesse et al., 2017). The LSTM baseline used the same hyperparameters, but with a learning rate of 1e-3 and value of n of 128 for the n-step TD updates. We found these values worked marginally better than other configurations for the LSTM.

### D.3 THE MINIGRID DOMAINS

The MiniGrid domains (Chevalier-Boisvert & Willems, 2018) consist of an agent (red triangle) in a grid environment. The agent can only see what is near to it, as shown in Figure 9. The environment contains many objects that the agent can interact with. The agent has 7 actions: turn left, turn right, move forward, pick up an object, drop the object being carried, toggle (open doors, interact with objects), and done (to complete a task). We ran experiments on the two environments that were designed to test the agent's memory capabilities: RedBlueDoors and MemoryS7.

Figure 9a shows the RedBlueDoors domain. In this environment, the agent is randomly located in a room with a red and a blue door. The agent is rewarded by opening the red door and then the blue door (in that order). Concretely, the agent receives a reward of 1 minus a small discount, which is proportional to the length of the episode, for solving this problem (and zero otherwise). The episode ends after opening the blue door or after reaching a time limit of 1280 steps.

Figure 9b shows the MemoryS7 domain. In this environment, the agent starts in a small room where it sees an object and then it has to navigate to the matching object at the other side of the room. Going to the right object results in a reward of 1. Going to the wrong object results in zero reward. The episode ends when the agent reaches any of the candidate objects or after reaching a time limit of 245 steps.

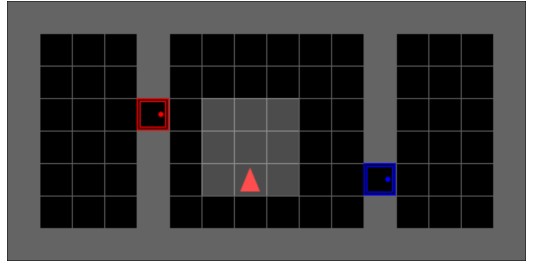

(a) RedBlueDoors domain.                    (b) MemoryS7 domain.

Figure 9: The minigrid environments.

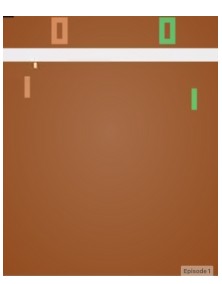       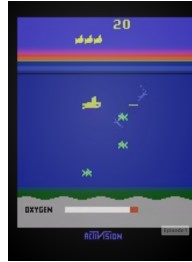

(a) Pong domain.          (b) Seaquest domain.

Figure 10: Atari environments.

The observations are encoded using one-hot representations. This encoding is a binary tensor that contains the objects in each tile that the agent can see. The objects are represented by three one-hot vectors that describe the object's type, color, and status (e.g., a closed red door).

We ran experiments using the OpenAI baseline implementation of PPO (Hesse et al., 2017). For O3, OA3, K3, and B3, the neural network used had 5 fully connected layers with 128 neurons per layer with *tanh* activation functions. The LSTM baseline used the same network with an extra LSTM layer of 128 neurons. All approaches were trained using the Adam optimizer (Kingma & Ba, 2014).

The approaches O3, OA3, K3, and B3 used a learning rate of 1e-5, a value of 128 for n in the n-step TD updates, 8 training minibatches per update, and 4 training epochs per update. The rest of the hyperparameters were set to their default values (Hesse et al., 2017). The LSTM baseline used the same hyperparameters, but a learning rate of 1e-3 in the MemoryS7 and 1e-5 in the Red-BlueDoors. Those learning rates improved the LSTM performance considerably. Note that these are the hyperparameters recommended by Willems (2019) to train PPO with LSTMs over MiniGrid environments. The only exception is the learning rate, which we fine-tuned.

## D.4 THE ATARI DOMAINS

Figure 10 shows snapshots of two atari games: Pong and Seaquest. The objective of Pong is to control the green paddle to hit the white ball and beat the opponent (red paddle) by getting the ball passed the opponent's paddle. The objective of Seaquest is to control a submarine to shoot at enemies and rescue divers. We ran experiments using the OpenAI's interface (Brockman et al., 2016) of the arcade learning environment (Bellemare et al., 2013). To incorporate some partial observability, we gave the agent only one frame as input. Hence, the agent cannot know the direction where objects are moving. Following Machado et al. (2018)'s recommendations, we made these environments stochastic by including sticky actions with 0.25 probability and forcing the agent to randomly execute up to 30 no-op actions at the beginning of each episode. We also used three standard practices introduced by DeepMind (Mnih et al., 2015) for training agents in atari domains. These are to reduce the input image to a grayscale image of 84x84 pixels, to clip the rewards, and to end the episode when the agent loses a life. In these domains, we used the same (convolutional) neural network and hyperparameters proposed by Hesse et al. (2017).

Table 1: Speedups using PPO. Each value shows the speedup relative to training an LSTM policy.

| Domain | Memory | CPU | GPU | Domain | Memory | CPU | GPU |
|--------|--------|-----|-----|--------|--------|-----|-----|
| Cookie | None | 10.87 | 2.32 | Keys | None | 7.40 | 3.45 |
| | K6 | 8.63 | 2.43 | | K6 | 5.13 | 2.97 |
| | B6 | 10.13 | 2.56 | | B6 | 5.86 | 3.09 |
| | O6 | 9.85 | 2.03 | | O6 | 5.37 | 2.93 |
| | OA6 | 8.24 | 1.92 | | OA6 | 4.32 | 2.75 |
| | LSTM | 1.00 | 1.00 | | LSTM | 1.00 | 1.00 |
| RBDoors | None | 2.99 | 3.15 | MemoryS7 | None | 2.91 | 2.24 |
| | K3 | 2.26 | 2.89 | | K3 | 2.73 | 2.78 |
| | B3 | 2.90 | 2.92 | | B3 | 3.01 | 2.13 |
| | O3 | 2.18 | 2.94 | | O3 | 2.80 | 2.07 |
| | OA3 | 2.48 | 2.34 | | OA3 | 2.40 | 2.76 |
| | LSTM | 1.00 | 1.00 | | LSTM | 1.00 | 1.00 |
| Pong | None | 1.51 | 2.18 | Seaquest | None | 1.40 | 2.72 |
| | K3 | 1.31 | 1.21 | | K3 | 1.26 | 2.11 |
| | B3 | 1.33 | 1.76 | | B3 | 1.19 | 1.63 |
| | O3 | 1.51 | 1.71 | | O3 | 1.06 | 2.11 |
| | OA3 | 0.57 | 0.67 | | OA3 | 0.21 | 0.40 |
| | LSTM | 1.00 | 1.00 | | LSTM | 1.00 | 1.00 |

### D.5 SPEEDUP COMPARISON

Learning memoryless policies for memory-augmented environments is usually faster than learning history-based policies. To show this, Table 1 reports the relative speedups of PPO using different forms of external memories with respect to PPO using an LSTM. To compute these values, we ran each agent for $200,000$ steps. In the CPU experiments, we used 16 logical cores from a AMD Ryzen Threadripper 2990WX processor. In the GPU experiments, we used one NVIDIA Tesla P100 GPU and 8 CPUs. In all these cases, learning a memoryless policy for a memory-augmented environment was faster than learning a history-based policy using an LSTM (sometimes over 10 times faster). The only exception was OA3 in Atari games – which was surprisingly slow.

### D.6 EXPERIMENTS USING TEMPORAL DIFFERENCE METHODS

As discussed in the paper, the POMDP theory suggests that policy learning methods should be preferred when facing memory-augmented environments. However, we have found that pure temporal difference methods can still take advantage of $Ok$ and $OAk$ memories. Here we show a few results using Sarsa($\lambda$) (Seijen & Sutton, 2014) and DDQN (Van Hasselt et al., 2016).

#### D.6.1 EXPERIMENTS USING SARSA($\lambda$)

This section discusses results using Sarsa($\lambda$) in the gravity domain. Sarsa($\lambda$) is a strong candidate to learn policies over memory-augmented environments because the value of $\lambda \in [0, 1]$ controls the degree in which this algorithm behaves as a pure TD method ($\lambda = 0$) or a pure Monte Carlo method ($\lambda = 1$). This is an important feature for memory-augmented environments because pure TD methods might be unable to accurately evaluate policies (see Section 5.1). In fact, Peshkin et al. (1999) recommended to use Sarsa($\lambda$) with a $\lambda$ close to 1 to learn policies over B$k$ memories.

In these experiments, we used $\epsilon$-greedy exploration (with $\epsilon = 0.01$), a discount factor of 0.95, and a learning rate of 0.1. We used an optimistic initialization for the q-values. Figure 11 shows the results for $\lambda$ equal 0, 0.5, and 1. Note that Sarsa($\lambda$) learns to control O1 and OA1 memories regardless of the value of $\lambda$. However, the most stable performance was obtained when using Sarsa(0.5) with O1. Note that Sarsa(1.0) was able to properly control the B1 memory, but its performance was unstable.

#### D.6.2 EXPERIMENTS USING DDQN

We also ran experiments using the OpenAI baseline implementation (Hesse et al., 2017) of DDQN (Van Hasselt et al., 2016). Figure 12 shows the results. Each line is the average reward per episode over 3 runs and the shadowed area represents half a standard deviation. In general, O$k$ memories

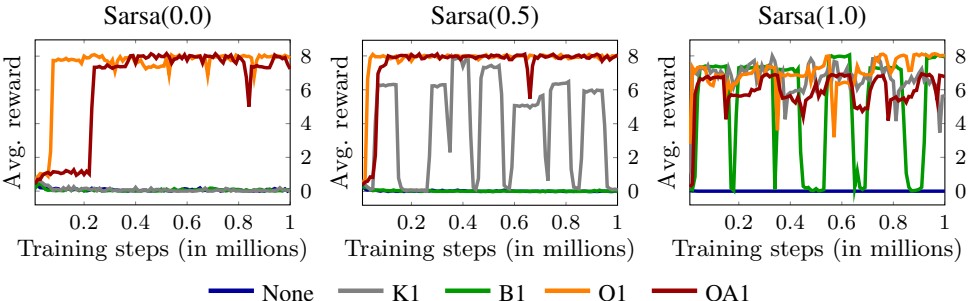

Figure 11: Tabular experiments in the gravity domain. We reported the avg. reward per 100 steps.

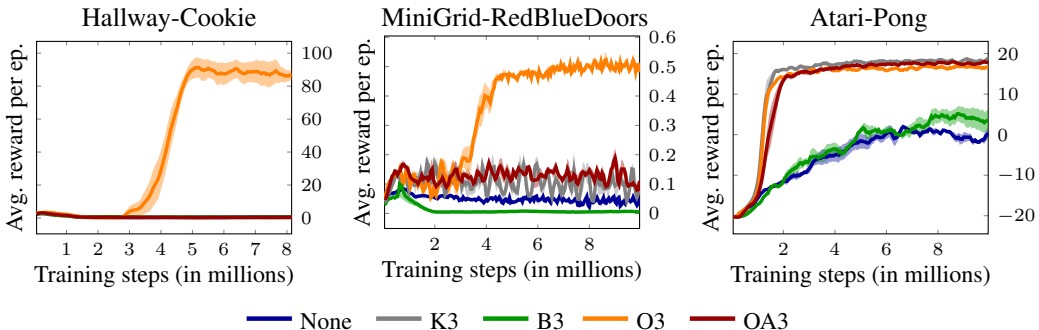

Figure 12: Results over partially observable benchmarks using DDQN and different memories.

outperform the other forms of memories, except for Pong where K3 performs the best (as expected). Note that OA3 performs well in Pong, while it did not when trained using PPO.

In the Cookie domain, the neural network used had 6 fully connected layers with 128 neurons per layer with *tanh* activation functions. On every step, we trained the network using 64 sampled experiences from a replay buffer of size $100,000$ using the Adam optimizer (Kingma & Ba, 2014), a discount factor of 0.99, and a learning rate of 1e-5. The exploratory policy was $\epsilon$-greedy with $\epsilon = 0.1$. The target network was updated every 100 steps.

In the RedBlueDoors domain, the neural network used had 4 fully connected layers with 128 neurons per layer with *tanh* activation functions. On every step, we trained the network using 32 sampled experiences from a replay buffer of size $100,000$ using the Adam optimizer (Kingma & Ba, 2014), a discount factor of 0.9, and a learning rate of 1e-5. The exploratory policy was $\epsilon$-greedy with $\epsilon = 0.1$. The target network was updated every 100 steps.

In Pong, we used the same (convolutional) neural network and hyperparameters proposed by Hesse et al. (2017) for DDQN.

