# OpenReview forum: "The act of remembering: A study in partially observable reinforcement learning"
_ICLR.cc/2021/Conference — Reject_

### Official Review · AnonReviewer1 · 2020-10-19
**Allowing a reinforcement learning agent to push its current observation into a k-sized queue**

**Rating:** 7
**Confidence:** 4

**Review:**

This paper focuses on reinforcement learning in partially-observable environments, and revisits the approach that consists of extending the agent with an external memory. The main contribution of the paper is the proposal (and evaluation) of adding an action to the agent, that allows it to push its current observation (and previous action is some cases) in a k-sized queue. The observation of the agent is extended with the contents of the queue. The main argument of the authors is that learning "when to push" is easier for the agent than learning "what to push" (as done with Neural Turing Machines and memory-bit external memories), and being able not to push the current observation allows the agent to remember past observations for longer durations, as opposed to k-step observation windows approaches.

The ideas proposed in the paper are simple and elegant. It seems surprising that this idea has not yet been proposed, but the paper shows great effort from the authors to look for related work, and discuss it. It just seems that a discussion of the work of McCallum in the late 90's is absent from this paper. McCallum, in his PhD thesis (and several papers before that), presents an algorithm that allows the agent to learn what observations to keep in a memory, and for how long. The implementation is, contrary to this paper, based on discrete data structures (trees for instance) and is more complicated than adding a "push current observation" action:

McCallum, R. "Reinforcement learning with selective perception and hidden state." (1997).

Work on Reinforcement Learning Neural Turing Machines is also related to what this paper present, but considers cases where the agent is trained (with backpropagation) on "what" to push on a memory tape, instead of simply being able to write the current observation on the memory tape:

Zaremba, Wojciech, and Ilya Sutskever. "Reinforcement learning neural turing machines-revised." arXiv preprint arXiv:1505.00521 (2015).

Clarity: my main complain with this paper is that it takes a very long time for the reader to see what the main contribution of the paper will be. The abstract and introduction both say that the paper will present a method that works in POMDPs, but does not state the method (the Ok memory). I would strongly advise the authors to explicitly mention, very early in the paper, that they propose to extend the agent with an action that allows the current observation to be enqueued in a fixed-length queue. A second (more minor) remark is that an intuition of what the Ok memory is (in addition to Definition 4.4) would help the reader understand that the agent observes the contents of the memory, and has an extra action to push the current observation onto it (is it really the case, by the way?)

Originality: the paper seems original. It is quite interesting that so much related work proposed ideas around what this paper proposes, but never exactly the formalism proposed in this paper.

Significance: the formalism presented in this paper could be applicable to a variety of POMDPs for which a finite set of past (possibly-continuous) observations help with executing good actions. Even though the paper does not point at real-world tasks that fit this definition, I believe that many "not that Markovian" real-world tasks could be solved using small Ok memories. As such, the significance of the work presented in this paper could be quite high.

To summarize my review:

- pros: simple idea, good empirical performance, applicable to many POMDPs
- cons: many pages of the paper must be read before understanding what its main contribution will be, and a discussion of industrial or real-world tasks that fit the class of POMDPs solvable with Ok memories would have been nice.

I therefore recommend (borderline) acceptance, but strongly suggest that the authors state their contribution in the abstract and introduction.

Author response: the updated paper is much clearer, and its abstract and introduction allow to clearly see what will be the contributions of this paper. I thank the authors for having followed my advice about this point. With this problem addressed, I recommend accepting this paper.

---

> ### Author Response · Authors · 2020-11-21
> **Response to R1**
>
> Thanks for your thoughtful and constructive review. We incorporated your suggestions into the new version of our paper. Concretely, we made the following changes to the paper that relate to your comments:
>
> 1) We elaborated on the details of our contribution in the abstract and introduction. (To answer your question, the Ok and OAk memories do indeed work just as you describe.)
>
> 2) We added a reference to McCallum’s thesis and discussion of RL-NTM (Zaremba et al., 2015) to the related work section. In the case of McCallum’s thesis, his tree-based approaches (in particular, USM and U-Tree) fit well in our discussion about model-based RL and we also mention them in Section 7 as part of a set of external memories that are worth studying in the context of memory-augmented environments.

---

> > ### Comment · AnonReviewer1 · 2020-11-23
> > **Very nice abstract!**
> >
> > I have looked at the revised paper, and it is indeed much clearer. I particularly like the abstract, and the explicit statement of the contributions at the end of the introduction (the fact that it is a list is quite in fashion currently, but the important part is that the "proposed method" is succinctly described somewhere early in the paper). The discussion of RL-NTM and McCallum's algorithms is good to me.

---

### Official Review · AnonReviewer4 · 2020-10-21
**Acting to remember, revisited**

**Rating:** 7
**Confidence:** 3

**Review:**

The paper extends the agent actions with an ability to write to an external memory. The paper does a nice survey of the previous approaches. The paper explains the difficulties with bootstrapping and policy improvement in POMDPs. The paper proposes simple memories for storing a buffer of k observations. The agent has the ability to push or not to push the current observation to the buffer. The whole content of the external memory is visible to the agent at each time step.

The paper is nicely written and clear. I enjoyed reading the survey of the related work and the explanation of the problems.
I welcome that the paper tries to use RL to control the external memory. This topic deserves more research.

Suggestions to increase the paper impact:
1) The survey of the existing literature can be made more valuable by mentioning RL methods suitable for function approximation. As explained in "Reinforcement Learning: An Introduction", Section 17.3 "Observations and State" by Sutton and Barto, partial observability is a special case of function approximation.
2) It would be nice to mention the limitations of the different memories. Maybe that would help you to design better memories.
For example, if the task requires to count to N, you would need a memory with log2(N) bits. You can also discuss the need to use long n-step returns, e.g., n=2048.
3) The proposed Ok, OAk memories are not better at all tasks than LSTM. Consider using these memories *together* with LSTM.
It would be nice to get the benefits of both approaches.

---

> ### Author Response · Authors · 2020-11-21
> **Response to R4**
>
> Thank you for your feedback. We also believe that this topic deserves more research and think that our work is an important step towards that end.
>
> We incorporated your suggestions in the new version of our paper. In Section 7, we note that theoretical guarantees for function approximation in RL also apply to partially observable RL.  Related to this discussion, we included a reference to the so-called non-delusional q-learning algorithm (Lu et al., 2018) which is guaranteed to converge to optimal memoryless policies in the limit, and as such relates to our memory-augmented environments.
>
> In Section 7 we additionally discuss the limitations of Ok and OAK memories resulting from the fixed size of their buffers and the additional constraint that the agent can only selectively push observations into that buffer. This restricts what can be remembered and learned. We discuss potential approaches to overcome those limitations, including, as you noted, learning an LSTM policy to control an Ok memory.

---

> > ### Comment · AnonReviewer4 · 2020-11-21
> > **Response to Authors**
> >
> > Thank you for your response and for incorporating the feedback.
> >
> > I like the paper, because it explains the general problems when trying to use RL methods to control an external memory. I'm less excited about the Ok and OAK memories. I consider them only as an example of additional memories controlled by RL. The paper should not oversell the Ok and OAK memories. You can avoid the following statement: "these memories outperformed k-order memories, binary memories, and LSTM memories in most of our experiments". The conclusion would be different, if using environments difficult for the Ok and OAK memories.

---

> > > ### Author Response · Authors · 2020-11-23
> > > **Response to R4**
> > >
> > > Thanks for the prompt feedback. We agree that Ok and OAk are only examples of memory forms that proved to be effective for the domains we experimented with; they are by no means the end of the story. They have their limitations, as discussed to some extent in Section 7. They are important in this paper in helping to establish the viability of the general approach of augmenting environments with structured memory and learning a policy that decides what to remember in each step. We aspire for this paper to encourage pursuit of this line of work. We will be more consistent in highlighting that we view Ok and OAk as a first step towards investigating various forms of memory (per Section 7), as a viable memory form for some problems, and as a future baseline.

---

### Official Review · AnonReviewer3 · 2020-10-28
**A simple fix for buffer-style memory systems with thorough theory and evaluation**

**Rating:** 6
**Confidence:** 3

**Review:**

This work proposes a lightweight approach to control memory in POMDPs. It is an alternative to heavier overhead approaches such as recurrent networks or memory-augmented networks (Oh et al. 2016).

The main contribution is to resurrect the old idea of simple lightweight memories and address what made them fail in complex domains by developing novel memory systems. The novel type of memory developed is the $Ok$ memory, wherein the agent is given a choice whether to push the most recent frame into the memory buffer or not. The commonly used $Kk$ memories in contrast push the most recent state into memory by default.

+ The paper develops the theory of memory augmented POMDPs from basic principles and formalizes the learning problem.
+ A simple idea is methodically and thoroughly explored through experimentation.
+ The paper is mostly well written and easy to follow with helpful toy problems and useful illustrations.

An obvious flaw with limited capacity buffer memory systems that store original observations is that longer term dependencies are harder to capture. This work offers a simple fix by including the decision of whether to store an observation or not into the agent's action space. But the system remains limited by having to store a fixed amount of full observations, as compared to more complex memory systems that can chose *what* to write along with when to write (eg. Neural map by Parisotto and Salakhutdinov 2017). Therefore, while the theory and evaluation are extensive, the memory system itself is limited and inefficient for environments where certain features of the state may need to be extracted and tracked for a long duration (not just a few frames).

---

> ### Author Response · Authors · 2020-11-21
> **Response to R3**
>
> Thank you for your review. We think it provides an accurate summary of our work and contributions. Motivated by your feedback, we added a section to the paper that discusses the limitations of Ok memories, and how to address them (see Section 7). We also added a citation to Parisotto and Salakhutdinov (2017), which was missing in our initial submission.
>
> As a brief summary of Section 7, in our paper we revisited an old idea for tackling partially observable RL. The idea was to provide an external memory to the RL agent where it could write information using actions under its control. However, that original idea was not working well in practice. As our paper shows, the problem was that previous works were using external memories that were too flexible (i.e., binary memories) which resulted in problems that were highly non-Markovian (see Section 5) and, as such, hard to solve using standard RL agents. Here we studied other forms of memories that buffered, rather than encoded, aspects of the state-action history. In this context, our Ok memories represent an important step towards studying various forms of memories that RL agents can actually exploit in practice. Their strong empirical performance (outperforming LSTM memories in our experiments), combined with their simplicity and ease of implementation, suggests that Ok memories have the potential to be broadly adopted. Further, they present a good baseline for measuring progress in partially observable RL going forward.
>
> That said, Ok memories are limited by the size of their buffers and by the fact that the agent can only push observations into that buffer (as you noted). Hence, there are problems that they cannot solve. For instance, Ok memories cannot keep track of whether the current time step is odd or even (which a B1 memory could do). We have added this to the discussion in Section 7. However, we believe that many of the limitations of Ok memories can be overcome by letting the agent decide on a position in the buffer to save (or remove) an observation or by training an LSTM policy to control an Ok memory.

---

### Official Review · AnonReviewer2 · 2020-10-29
**The notation and explanation needs some treatment. The contribution is empirical, comparison with stronger baseline seems missing.**

**Rating:** 5
**Confidence:** 3

**Review:**

In this paper, the authors propose a series of methods to tackle policy learning in POMDPs.

At the core of the proposed method sits an augmented memory that the agent is allowed to write on and read from at the time of decision making.

This allows the agent to store some of its past to be used for future decision making.

The authors raise a few training, stability, and limitation issues in prior work. And argue that their proposed method improves over the prior works.

The idea of using augmented memory is exciting and fundamental and definitely worth expanding.

However, I found a few issues with the presentation and contribution of this paper that I would be happy to share.

1) Second line of abstract:
"Learn- ing memoryless policies is efficient and optimal in fully observable environments." It is not clear what the authors mean here. Is the learning of memoryless policy efficient and optimal? If yes, it would great if the authors could parse it.

If they mean memoryless policies are optimal, then I recommend the authors to restate this statement since it is incorrect.

2) The authors state "can solve problems that were unsolvable using LSTMs" since it is an impossibility statement, I would recommend either proving a reference for such a statement or provide proof.

3) "Notice that neither q-learning nor 5-step actor-critic were able to understand how to use the B1 memory to consistently solve the gravity domain." I guess the authors mean the agent using q-learning or 5-step actor-critic were not able to learn how to use ... .

4) I strongly recommend the authors to use more concrete notation. It seems that they study episodic POMDP or maybe a fixed horizon. They mentioned episodic but did not define it. Also, it is not clear what are state, action, and observation spaces. Are they finite? In the analysis I found in the appendix, it seems the authors approach finite ones. But it would be useful to mention it in the main body since there are high-dimensional exps in the paper. Also, q is not defined. I checked the referenced paper,  Jaakkola et al. 1995, there it was also not clear what is q. They first define it for time step zero. Then later use it for any time step. They seem to not define P_\pi(s|m). It would be great if the authors could define these quantities. The authors state that "Pπ (s|o) is the probability of being in state s given that the observation is o, when following policy π" well, at what time step? Please define these terms.

5) I did not understand the point of a recall task example on page 5.
As the authors know, there are examples of POMDPs that constant actions are optimal. I am not sure what would a significant conclusion one can draw from such examples.


----------------
General evaluation:
6) Almost all the theoretical statements are straightforward results of definitions and provided for justification. They are useful in understanding the paper. I appreciate the authors for including them.

7) I again encourage the authors to make their notation a bit more concrete. If they study fixed horizon POMDPs, where the stages (time step in the episode) are encoded?

8) While I appreciate the augmented memory type policy, it seems the authors' proposed method is quite fragile. For example, for the OAk setting, if the agent needs to store k pairs of (o_t,a_{t-1}) to achieve a good solution, then the method breaks?

9) Such fragileness mentioned in 8 seems not to be an issue in methods that learn latent states. How would the authors handle that?

10) Since the contribution is empirical, I would be happy if the authors provide a study against existing baselines, e.g., those referred to in the related works.

11) Regarding memoryless policies in pomdps, I recommend the authors to take a look at Policy Gradient in Partially Observable Environments: Approximation and Convergence. They seem to have some convergence analysis that might be useful.



---------
Post rebuttal.
I would like to thank the authors for their clarifying reply. I will discuss with other reviewers and AC, and update my evaluation further.

---

> ### Author Response · Authors · 2020-11-21
> **Response to R2 (part 1)**
>
> Thank you for the constructive feedback. It helped us improve the clarity of our work. We have uploaded a revised version of the paper and added some further clarifications below. We welcome further questions or comments.
>
> > In this paper, the authors propose a series of methods to tackle policy learning in POMDPs. At the core of the proposed method sits an augmented memory that the agent is allowed to write on and read from at the time of decision making. This allows the agent to store some of its past to be used for future decision making. The authors raise a few training, stability, and limitation issues in prior work. And argue that their proposed method improves over the prior works. The idea of using augmented memory is exciting and fundamental and definitely worth expanding. However, I found a few issues with the presentation and contribution of this paper that I would be happy to share.
>
> **Response**: Thanks. We also think that this is an exciting and fundamental idea. Your feedback was divided thematically into  “presentation” issues and “general evaluation” issues.  We address each below, in turn.
>
> **Presentation**:
>
> > 1. Second line of abstract: "Learning memoryless policies is efficient and optimal in fully observable environments." It is not clear what the authors mean here. Is the learning of memoryless policy efficient and optimal? If yes, it would great if the authors could parse it. If they mean memoryless policies are optimal, then I recommend the authors to restate this statement since it is incorrect.
>
> **Response**: We replaced the statement of concern with the following: “In fully observable environments it is sufficient for RL agents to learn memoryless policies.” This is predicated on the observation that in a fully-observable environment, such as an MDP, any optimal memoryless policy is as good as any optimal history-based policy. As such, memoryless policies are all you need in fully observable problems.
>
> > 2. The authors state "can solve problems that were unsolvable using LSTMs" since it is an impossibility statement, I would recommend either proving a reference for such a statement or provide proof.
>
> **Response**: We removed the concerning statement from the introduction, which we agree was misleading. The statement was not intended to be an impossibility statement. Rather it was alluding to the fact that no single run of PPO using an LSTM memory was able to solve two of our domains (the hallway domains, in particular), performing no better than a random policy.
>
> > 3. "Notice that neither q-learning nor 5-step actor-critic were able to understand how to use the B1 memory to consistently solve the gravity domain." I guess the authors mean the agent using q-learning or 5-step actor-critic were not able to learn how to use ... .
>
> **Response**: We fixed this.
>
> > 4. I strongly recommend the authors to use more concrete notation. It seems that they study episodic POMDP or maybe a fixed horizon. They mentioned episodic but did not define it. Also, it is not clear what are state, action, and observation spaces. Are they finite? In the analysis I found in the appendix, it seems the authors approach finite ones. But it would be useful to mention it in the main body since there are high-dimensional exps in the paper. Also, q is not defined. I checked the referenced paper, Jaakkola et al. 1995, there it was also not clear what is q. They first define it for time step zero. Then later use it for any time step. They seem to not define $P_{\pi}(s|m)$. It would be great if the authors could define these quantities. The authors state that "Pπ (s|o) is the probability of being in state s given that the observation is o, when following policy π" well, at what time step? Please define these terms.
>
> **Response**: Our theoretical analysis is for finite POMDPs. Thus, the number of states, actions, and observations are finite. Beyond that, there is no theoretical restriction regarding whether the POMDP is episodic, or whether it has a finite or infinite horizon.  Hence, our definitions of MDPs and POMDPs in Section 2 correspond to finite MDPs (resp. POMDPs). That said, our approach can be used with any RL agent and, as such, can also solve POMDPs with continuous observations and actions  -- as shown in our experiments. Following your suggestion, we added the mathematical definitions of the q-value functions $q(s,a)$ and $q(o,a)$ to the paper (see Sections 2 and 5.1). Note that they are defined for any time step (as is the norm). Similarly, $P_{\pi}(s|o)$ is also defined for any time step. We introduce $P_{\pi}(s|o)$ and clarifying that $P_{\pi}(s|o)$ is defined for any time step. Since this notation is introduced for the purpose of providing intuition regarding theoretical results from the work of Jaakkola et al. (1995), we felt the treatment was sufficient and do not elaborate further.

---

> > ### Author Response · Authors · 2020-11-21
> > **Response to R2 (part 2)**
> >
> > > 5. I did not understand the point of a recall task example on page 5. As the authors know, there are examples of POMDPs that constant actions are optimal. I am not sure what would a significant conclusion one can draw from such examples.
> >
> > **Response**: We use the recall task as a handy running example to explain some of the key concepts that play a role in learning memoryless policies in memory-augmented environments. We also use the recall task to prove Theorem C.6.
> >
> >
> > **General evaluation**:
> >
> > > 6. Almost all the theoretical statements are straightforward results of definitions and provided for justification. They are useful in understanding the paper. I appreciate the authors for including them.
> >
> > **Response**: Thanks!
> >
> > > 7. I again encourage the authors to make their notation a bit more concrete. If they study fixed horizon POMDPs, where the stages (time step in the episode) are encoded?
> >
> > **Response**: Thanks. We clarified the notation per question 4 above. (If concerns remain, please let us know.) In reference to this question, we did not study fixed horizon POMDPs and the time steps were not included as part of the observations given to the agent.
> >
> > > 8. While I appreciate the augmented memory type policy, it seems the authors' proposed method is quite fragile. For example, for the OAk setting, if the agent needs to store k pairs of (o_t,a_{t-1}) to achieve a good solution, then the method breaks?
> >
> > **Response**: To put this in context, the idea of providing external memories to RL agents (and actions to control it) has been around since the 90s, but it was not working. Part of the problem was that previous works were using binary memories -- which are so flexible that it is hard for RL agents to understand how to use them (as discussed in Sections 4 and 5). Our OAk memories are more constrained but have stronger empirical performance. As such, we see OAk as an important step towards understanding which types of external memories RL agents can learn to control in service of solving partially observable problems. To your question, note that we include a new section in the paper (Section 7) which discusses the limitations of OAk memories and how to address them.
> >
> > > 9. Such fragileness mentioned in 8 seems not to be an issue in methods that learn latent states. How would the authors handle that?
> >
> > **Response**: Solving this problem is relatively easy. You could learn an LSTM policy that also controls an OAk memory. In general, LSTM policies excel at combining recent information into their latent state, but they fail at remembering information that is far in the past (which is basically why they perform poorly in some of our domains). In this respect, the OAk memory could help the LSTM to remember important events that are far in the past in order to inform future actions. We mention this idea in Section 7.
> >
> > > 10. Since the contribution is empirical, I would be happy if the authors provide a study against existing baselines, e.g., those referred to in the related works.
> >
> > **Response**: We do compare against three important baselines. These are k-order memories (Mnih et al., 2015), binary memories (Peshkin et al., 1999), and LSTM memories (Hausknecht& Stone, 2015). The comparison with binary memories is important because they are the original form of external memory proposed for training agents that attempt to solve POMDPs by learning both how to act and what to remember. The comparison with k-order and LSTM memories is important because these two memories are the most commonly used approaches to tackle partially observable RL; they are simple to implement and perform well in practice. As such, the fact that Ok and OAk memories outperform k-order and LSTM memories in our experiments, and that these memories are as simple to implement as k-order memory, suggests that our method could become a widely used approach for partially observable RL. Finally, note that our contribution is not only empirical. We also formally defined memory-augmented environments and proved several properties for them in the appendix, e.g., sufficient conditions for expressiveness of Bk, OAk, Ok (Proposition A.2) and that the algorithm of Jaakkola, even with a sufficiently expressive external memory, will not converge to an optimal memoryless policy for some POMDPs (Theorem C.6).
> >
> > > 11. Regarding memoryless policies in pomdps, I recommend the authors to take a look at Policy Gradient in Partially Observable Environments: Approximation and Convergence. They seem to have some convergence analysis that might be useful.
> >
> > **Response**: Thanks, we added that reference to the paper.

---

### Author Response · Authors · 2020-11-21
**A revised version of our paper is now available**

Thank you for all your constructive feedback. We just uploaded a revised version of our work. This new version incorporates most of your suggestions and we highlighted in blue the most notable changes. Please let us know if you have further feedback.

---

### Decision · Program_Chairs · 2021-01-07
**Final Decision**

**Decision:**

Reject

**Comment:**

This paper presents a refreshening insight into the classical idea of using external memory for reinforcement learning agents that learn and act in partially observable environments. The authors investigate a number of different memory architectures (Ok, OAk, Kk) and provide an insightful discussion on why we want to restrict the structure of the memory.

Reviewers generally appreciated the technical contribution of the paper, although not very convinced that this work will have a significant impact on future work. AC is also not sure about the conclusion drawn from the paper, where policies with external memory could have better sample complexity compared to rnn-based policies. BTTT is computationally expensive, but it shall give better direction of which state to jump to, compared to the authors approach where the gradients are stopped at every timestep. So there should be pros and cons about this approach, and AC suspects that the sample complexity improvement actually comes from the fact that authors are explicitly limiting what can be stored in the memory, e.g. O or OA. This advantage can be broken in some other domains. AC admits that this is only a speculation at this point, but the motivation to use the external memory framework proposed in the paper needs to be more carefully investigated.